# Filopodia-like protrusions of adjacent somatic cells shape the developmental potential of oocytes

Flora Crozet[1,2], Gaëlle Letort[1,2], Rose Bulteau[1], Christelle Da Silva[1], Adrien Eichmuller[1,3], Anna Francesca Tortorelli[3], Joséphine Blévinal[4], Morgane Belle[4], Julien Dumont[1], Tristan Piolot[1], Aurélien Dauphin[3], Fanny Coulpier[5], Alain Chédotal[4], Jean-Léon Maître[3], Marie-Hélène Verlhac[1], Hugh J Clarke[6], Marie-Emilie Terret[1]

The oocyte must grow and mature before fertilization, thanks to a close dialogue with the somatic cells that surround it. Part of this communication is through filopodia-like protrusions, called transzonal projections (TZPs), sent by the somatic cells to the oocyte membrane. To investigate the contribution of TZPs to oocyte quality, we impaired their structure by generating a full knockout mouse of the TZP structural component myosin-X (MYO10). Using spinning disk and super-resolution microscopy combined with a machine-learning approach to phenotype oocyte morphology, we show that the lack of *Myo10* decreases TZP density during oocyte growth. Reduction in TZPs does not prevent oocyte growth but impairs oocyte-matrix integrity. Importantly, we reveal by transcriptomic analysis that gene expression is altered in TZP-deprived oocytes and that oocyte maturation and subsequent early embryonic development are partially affected, effectively reducing mouse fertility. We propose that TZPs play a role in the structural integrity of the germline–somatic complex, which is essential for regulating gene expression in the oocyte and thus its developmental potential.

## Introduction

During oogenesis, mammalian oocytes undergo a process of differentiation that determines their quality and thus their developmental potential as early embryos after fertilization. Oocyte differentiation begins with the entry into a growth phase during which the oocyte increases in size and stores large amounts of macromolecules necessary to complete its development and early embryogenesis. During growth, the oocyte remains arrested in prophase and gradually acquires the competence to resume meiosis (1). Once fully grown, it further differentiates by undergoing meiotic maturation. During maturation, the oocyte completes the first meiotic division (meiosis I) and becomes arrested at metaphase of the second meiotic division (meiosis II). Meiosis completion then occurs if fertilization takes place.

The oocyte does not progressively acquire meiotic and developmental competence by itself in an autonomous way, but through a close dialogue with the somatic cells that surround it, called follicular cells, which themselves undergo a parallel and intricate process of differentiation (2, 3, 4, 5). Together with the oocyte, the follicular cells form a germ–somatic complex called the ovarian follicle.

Although the oocyte communicates with its surroundings via secreted factors (2, 4, 6), somatic communication occurs mostly through direct cell–cell contacts mediated by actin-rich filopodia-like (7) protrusions called transzonal projections (TZPs) (6, 8, 9, 10). These TZPs cross the *zona pellucida*, the extracellular matrix surrounding the oocyte, and contact the oocyte membrane at their tips. They allow follicular cells to metabolically support the oocyte and prevent early meiotic resumption (3, 6). In addition, TZPs may provide structural support in the follicle by maintaining cohesion between the oocyte and follicular cells (9, 11).

Although the dialogue within the ovarian follicle is increasingly emerging as a major player in the emergence of oocyte quality, the role of direct cell–cell contact communication between the oocyte and follicular cells is still poorly understood and may shed additional light on the acquisition of oocyte developmental competence. In this study, we sought to characterize the contribution of TZPs to oocyte development by impairing the structure of TZPs. As a candidate gene, we selected myosin-X (MYO10), a structural component of TZPs (7) known to promote filopodium formation (12, 13, 14) and potentially involved in the formation or maintenance of TZPs (15). We generated a full knockout mouse of *Myo10* in all cell types to completely deplete it from the ovarian follicle (*Myo10⁻/⁻* full). Furthermore, we took advantage of our previously generated mouse strain conditionally deleted for *Myo10* specifically in oocytes from early growth onward (*Myo10⁻/⁻* oo) (16) to help distinguishing phenotypes caused by loss of somatic MYO10 or loss of oocyte

[1]Center for Interdisciplinary Research in Biology, Collège de France, CNRS, INSERM, Université PSL, Paris, France   [2]Department of Developmental and Stem Cell Biology, Institut Pasteur, CNRS UMR 3738, Université Paris Cité, Paris, France   [3]Institut Curie, PSL Research University, Sorbonne Université, CNRS UMR 3215, INSERM U934, Paris, France   [4]Institut de la Vision, UMRS968/UMR7210/UM80, Paris, France   [5]Genomics Core Facility, Institut de Biologie de l'ENS, Département de biologie, Ecole normale supérieure, CNRS, INSERM, Université PSL, Paris, France   [6]Department of Obstetrics and Gynecology, McGill University, Montreal, Canada

Correspondence: marie-emilie.terret@college-de-france.fr

MYO10. We show that the loss of somatic but not oocyte MYO10 greatly decreases the density of TZPs. Surprisingly, oocytes deprived of TZPs develop to a normal size, but display oocyte-matrix defects. Importantly, gene expression is altered in TZP-deprived oocytes, correlating with lower rates of oocyte maturation and subsequent early embryonic development, inducing lower fertility in mice.

# Results

## Global deletion of *Myo10* decreases the density of TZPs without altering ovarian follicular organization

To deplete TZPs, we generated full knockout mice lacking the TZP component MYO10 in all cell types (Fig S1A–D). Loss of MYO10 expression in mice was previously characterized as semi-lethal, with some animals dying during embryogenesis whereas others being able to develop until adulthood (14). Similar to this study, we obtained adult $Myo10^{-/-}$ mice (named $Myo10^{-/-}$ full thereafter) and most showed phenotypes characteristic of the earlier described *Myo10* full knockout, such as the presence of a white spot on the abdomen and webbed fingers (14) (Fig S1B). In addition to this strain, we used our previously generated mouse strain conditionally deleted for *Myo10* in oocytes from early growth onward (named $Myo10^{-/-}$ oo thereafter) as a control to potentially distinguish oocyte phenotypes related to germline loss of MYO10 versus those to somatic loss of MYO10 (16) (Fig S1A and D).

In control follicles, MYO10 is found diffuse-like in the cytoplasm of the oocyte and of follicular cells (16) (Figs 1A and S1D) and accumulates into foci within the *zona pellucida* where TZPs are located (7, 15) (Fig 1A and B). These foci were entirely lost from the *zona pellucida* of oocytes coming from $Myo10^{-/-}$ full follicles (Fig 1A and B, upper panels). Conversely, the foci were present in the *zona pellucida* coming from $Myo10^{-/-}$ oo follicles lacking MYO10 only in the oocyte (Fig 1A and B, lower panels), indicating that MYO10 foci in the *zona pellucida* originate from follicular cells. Importantly, the loss of MYO10 foci in the *zona pellucida* of oocytes coming from $Myo10^{-/-}$ full follicles was concomitant with a significant decrease in the density of TZPs (Fig 1A and B, upper panels). This decrease was most apparent when imaging TZPs by super-resolution microscopy in $Myo10^{-/-}$ full oocytes cleared of follicular cells (Video 1, OMX; Fig 1C, STED). This reduction was not observed in oocytes coming from $Myo10^{-/-}$ oo follicles (Fig 1A and B, lower panels), implying that TZP density is related to the presence of MYO10 in follicular cells.

Consistent with these observations in fixed follicles, we also measured a fivefold decrease in TZP density in live, fully grown $Myo10^{-/-}$ full oocytes stained with SiR-actin to label F-actin (Fig 1D, quantification in E). Conversely, TZP density remained similar to controls in live $Myo10^{-/-}$ oo oocytes labeled with SiR-actin (Fig 1D and E), indicating that $Myo10^{-/-}$ oo oocytes retain a canonical density of follicular cell contacts. Because a few TZPs in the ovarian follicle do not contain actin (7, 10), we confirmed that loss of MYO10 decreases not only actin-rich TZPs but also total TZP density by staining oocytes with the membrane probe FM 1–43 (Fig S2A). We then wanted to determine when this phenotype of decreased TZP density appeared during follicular growth. Follicles at an early stage of growth contain few TZPs (7), so it was difficult to determine

whether there is a decrease in TZP density in $Myo10^{-/-}$ full oocytes at the beginning of growth compared with controls. However, the phenotype of decreased TZP density in the absence of MYO10 was apparent as early as mid-growth (15) (Fig S1D, right panel, magnifications), and TZP density was low in oocytes from $Myo10^{-/-}$ full follicles at the end of growth regardless of the oocyte diameter (Fig S2B). We next wondered whether and to what extent cumulus–oocyte coupling was affected in *Myo10* full knockouts. To assess coupling, we performed dye diffusion assays, that is, microinjection of a fluorophore (Lucifer Yellow) into the oocyte to visualize whether it could spread into the surrounding somatic cells (Fig S2C) as in Reference 7. As a negative control, we treated control complexes with carbenoxolone (CBX), a gap junction blocker, to prevent the transfer of Lucifer Yellow from the oocyte to the surrounding somatic cells via gap junctions (Fig S2C) as in Reference 7. Our results show that overall, some coupling remains in *Myo10* full knockouts, compared with control complexes treated with CBX where coupling is abolished (Fig S2C). We then quantified the coupling between the oocyte and follicular cells. For this, we measured the ratio of Lucifer Yellow fluorescence intensity between follicular cells and the oocyte 30 min after Lucifer Yellow microinjection into the oocyte. Measurements were performed on control cumulus–oocyte complexes, *Myo10* full knockout complexes, and control complexes treated with CBX when single follicular cells were visible, still connected to the oocyte by TZPs (visible with SiR-actin). Our results show that the amount of coupling between the oocyte and follicular cells is comparable between control and *Myo10* full knockouts, and the coupling is abolished by blocking gap junction communication with CBX (Fig S2D). Thus, *Myo10* full knockouts have fewer TZPs, but their remaining TZPs are functional for intercellular coupling, allowing exchanges between the oocyte and follicular cells, even if globally reduced.

Impaired intercellular communication between the oocyte and follicular cells in mice deficient for the gap junction subunit connexin 37 (17) or deficient for the oocyte-secreted factor GDF9 (18, 19) led to ovaries with abnormal morphology, lacking fully grown follicles. We therefore tested whether decreasing TZP density altered the integrity of $Myo10^{-/-}$ full ovaries. To gain access to intra-ovarian organization, whole ovaries from adult mice were transparized (20). As in control ovaries, $Myo10^{-/-}$ full ovaries contained late antral follicles (Fig S3A). $Myo10^{-/-}$ full mice also showed a variation in the volume of their ovaries (Video 2 and Video 3). Because of this variation, we assessed global ovarian integrity by measuring the follicular density of the ovary (number of follicles normalized by the ovarian area) rather than the absolute number of follicles per ovary. Follicular density was not significantly different between control and $Myo10^{-/-}$ full ovaries (Fig S3B), indicating that the complete loss of MYO10 and subsequent reduction in TZP density do not alter global follicular organization in the ovary. Thus, we confirmed that MYO10 is a structural component of TZPs, essential for TZP formation or maintenance (7, 15).

## Oocytes deprived of TZPs reach a canonical size at the end of growth but are morphologically different from controls

To determine the TZP-mediated contribution of follicular cells to oocyte development, we assessed whether $Myo10^{-/-}$ full oocytes

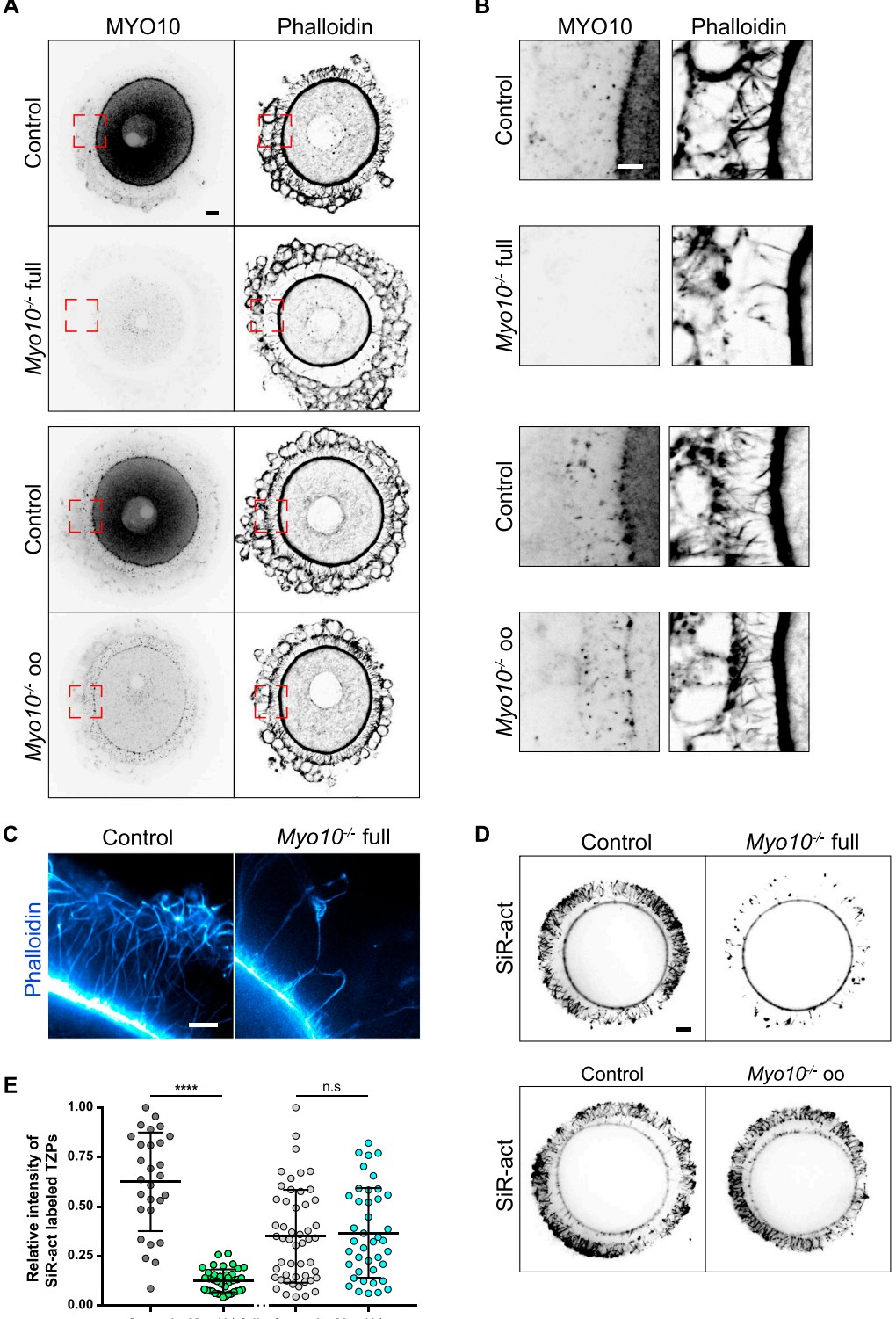

**Figure 1. Global deletion of *Myo10* decreases the density of transzonal projections.**
**(A)** Cumulus–oocyte complexes stained for myosin-X (MYO10, left images) and with phalloidin to label F-actin (right images). The upper panel shows complexes from the *Myo10* full knockout strain (full), and the lower panel, complexes from the *Myo10* oocyte knockout strain (oo). For each panel, control complexes are at the top, and *Myo10*⁻/⁻ complexes, at the bottom. For MYO10 staining, contrast adjustment is similar between complexes of the same strain. Scale bar = 10 µm. **(A, B)** Cropped images of the red dotted rectangles shown in (A) focusing on the *zona pellucida*. MYO10 staining is on the left, and phalloidin, on the right. Scale bar = 5 µm. **(C)** STED microscopy images of phalloidin-labeled oocytes arrested in prophase freed of follicular cells, focusing on the *zona pellucida* of a control oocyte (left) and a *Myo10*⁻/⁻ full oocyte (right). Scale bar = 2 µm. **(D)** Live fully grown oocytes arrested in prophase stained with SiR-actin to label F-actin. The upper images are oocytes from the *Myo10* full knockout strain (full), and the lower images are those from *Myo10* oocyte knockout strain (oo). Controls are on the left, and *Myo10*⁻/⁻ oocytes, on the right. Scale bar = 10 µm. **(E)** Scatter plot of the intensity of all SiR-actin–labeled TZPs of fully grown oocytes arrested in prophase. Control and *Myo10*⁻/⁻ full oocytes are in dark gray and green, respectively. Control and *Myo10*⁻/⁻ oo oocytes are in light gray and blue, respectively. (n) is the number of oocytes analyzed. Data are the mean ± s.d. with individual data points plotted. Data are from three to five independent experiments. Statistical significance of differences was assessed by an ANOVA or a Kruskal–Wallis test depending on whether the data followed a Gaussian distribution, $P < 0.0001$ (****, Ctrl full versus *Myo10*⁻/⁻ full), $P = 0.6857$ (n.s, Ctrl full versus Ctrl oo), $P = 0.4479$ (n.s. Ctrl full versus *Myo10*⁻/⁻ oo), $P < 0.0001$ (****, *Myo10*⁻/⁻ full versus Ctrl oo), $P < 0.0001$ (****, *Myo10*⁻/⁻ full versus *Myo10*⁻/⁻ oo), and $P = 0.9722$ (n.s, Ctrl oo versus *Myo10*⁻/⁻ oo). n.s, not significant.

deprived of TZPs developed properly. We used our previously developed deep-learning approach to characterize in an automatic manner the morphology of *Myo10*⁻/⁻ full oocytes at the end of the growth phase with the software Oocytor (21) to assess their meiotic competence (defined as the ability of oocytes to resume meiosis at the end of growth and mature properly). First, *Myo10*⁻/⁻ full oocytes grew to normal sizes (Fig 2A and B) and were arrested in prophase. We then measured the thickness and the perimeter of the *zona*

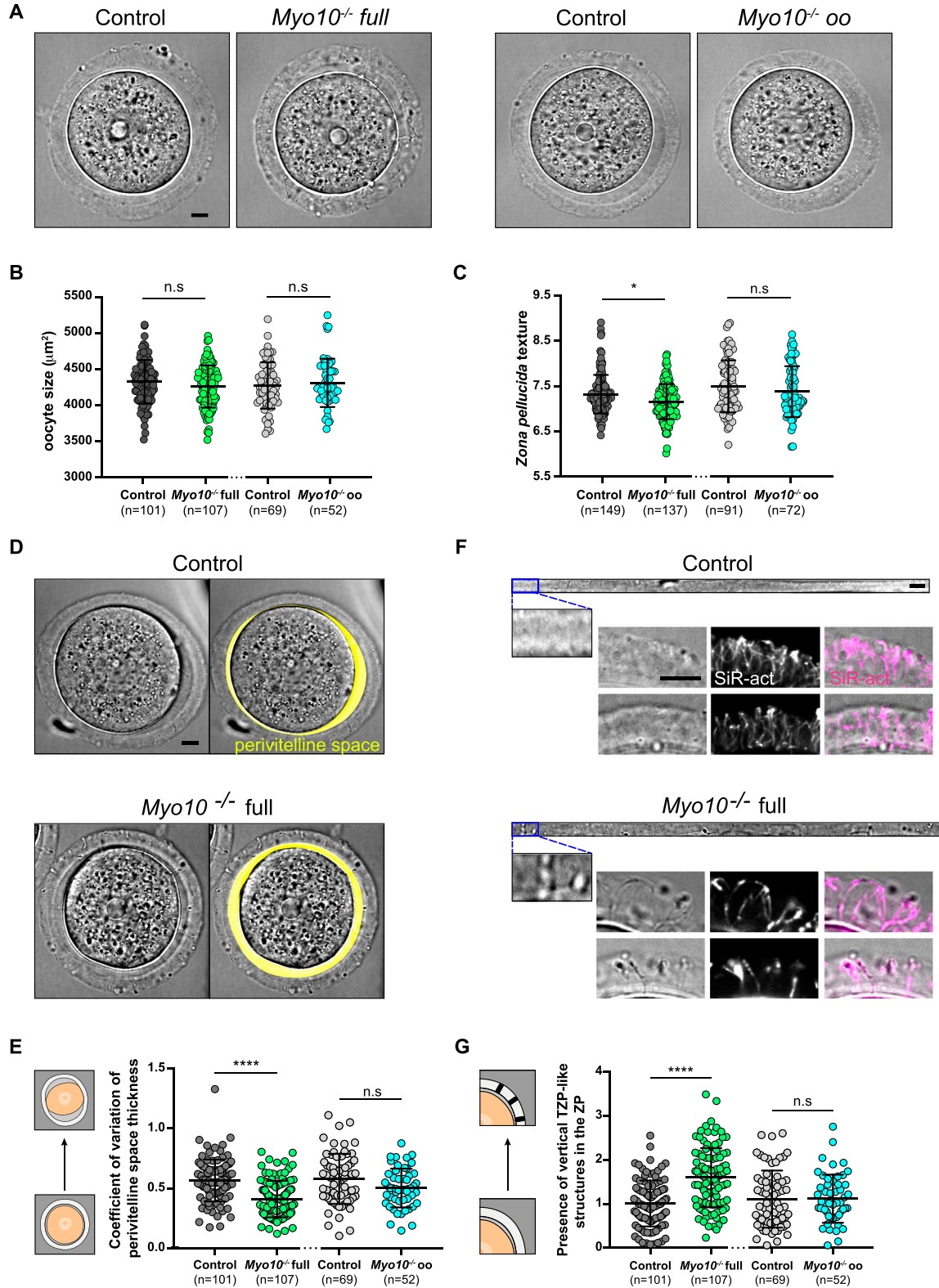

**Figure 2. TZP-deprived oocytes have morphological alterations.**
**(A)** Brightfield images of fully grown oocytes from the *Myo10* full knockout strain (left panel) and the *Myo10* oocyte knockout strain (right panel). For each panel, controls are on the left, and *Myo10*$^{-/-}$ oocytes, on the right. Scale bar = 10 $\mu$m. **(B)** Scatter plot of the equatorial plane area of control and *Myo10*$^{-/-}$ full oocytes (in dark gray and green, respectively) and control and *Myo10*$^{-/-}$ oo oocytes (in light gray and blue, respectively). (n) is the number of oocytes analyzed. Data are the mean ± s.d. with individual data points plotted. Data are from four to 13 independent experiments. Statistical significance of differences was assessed by an ANOVA or a Kruskal–Wallis

*pellucida*, features that reflect the ability of the oocyte to resume meiosis (21), and showed that they are similar between $Myo10^{-/-}$ full oocytes and controls (Fig S4A and B). However, the texture of the *zona pellucida*, one of the two most important features to assess oocyte maturation potential (21), was lower in $Myo10^{-/-}$ full oocytes (Fig 2C). Thus, the morphology of $Myo10^{-/-}$ full oocytes at the end of growth suggests that they have a similar potential for meiosis resumption as controls, but a decreased potential for oocyte maturation.

We further explored the differences in morphology between $Myo10^{-/-}$ full and control fully grown oocytes using our machine-learning algorithm (21) (Figs 2D–G and S4D–G), allowing us to detect in an unbiased manner the morphological features that differed the most between control and $Myo10^{-/-}$ full oocytes. One of the most important features used by our algorithm to distinguish control from $Myo10^{-/-}$ full oocytes was the composition of the *zona pellucida* with the presence of vertical structures resembling TZPs crossing the *zona pellucida* (Figs 2D, F, and G and S4D and G). Intriguingly, these structures were detected more frequently in $Myo10^{-/-}$ full oocytes than in controls (Fig 2G), despite the fact that total TZP density is decreased in $Myo10^{-/-}$ full oocytes. These structures, visible in transmitted light in *Myo10* full knockouts, always contain actin (Fig 2F). This implies that even though there are less TZPs in *Myo10* full knockouts (Fig 1E), they are more visible by transmitted light than those in controls, which are more abundant (Fig 1E) but less visible by transmitted light (Fig 2F). Our observations suggest that in *Myo10* full knockouts, the structure of the remaining TZPs and/or that of the *zona pellucida* is different from that of control oocytes. Interestingly, we occasionally observe some follicular-like cells located ectopically within the perivitelline space (the space between the oocyte and the *zona pellucida*) of *Myo10* full knockouts (Fig S4C), and the texture of the *zona pellucida* is different between control and *Myo10* full knockouts (Fig 2C). We therefore hypothesize that the structure of the *zona pellucida* (a viscoelastic extracellular matrix) may change as a function of the cross-linking generated by the number of TZPs passing through it. If so, this could change its mechanical properties. We thus assessed the mechanical properties of the *zona pellucida* of control and

*Myo10* full knockouts using atomic force microscopy (AFM) as in Reference 22. Our experiments show decreased *zona pellucida* elasticity in *Myo10* full knockouts compared with controls (Fig S4H). This suggests that the reduced number of TZPs could affect the structure, maybe via reduced cross-linking of the *zona pellucida*, and thus the mechanical properties of the *zona pellucida*.

The second important feature used by our algorithm was the distribution of the perivitelline space around the oocyte, which was detected to be more uniform in $Myo10^{-/-}$ full oocytes than in controls (Fig 2D, quantification in E; Fig S4D and G), as if oocytes lacking TZPs were more homogeneously detached from the *zona pellucida*. Furthermore, our machine-learning algorithm identified $Myo10^{-/-}$ full oocytes as more circular than control oocytes (Fig S4D, E, and G), as indicated by a lower aspect ratio in TZP-deprived oocytes (Fig S4E). In accordance with this feature, the perimeter of $Myo10^{-/-}$ full oocytes was slightly smaller than control oocytes for a similar area (Fig S4F).

Thus, using automatic approaches, we show that $Myo10^{-/-}$ full oocytes can reach normal sizes but display morphological differences from controls, in particular regarding the *zona pellucida* texture, a feature known to predict meiosis entry and maturation outcome (21).

### Gene expression is modified in oocytes deprived of TZPs

We next wondered whether the differences both in the density of communicating structures between the oocyte and its follicular cells and in oocyte morphology translated into differences in oocyte developmental potential. The oocyte acquires its developmental potential during its growth, where it accumulates a large number of transcripts. Transcription decreases dramatically at the end of oocyte growth and increases after fertilization in the zygote (23); thus, protein synthesis during the meiotic divisions relies predominantly on the translational regulation of transcripts accumulated during growth, making this storage critical for oocyte quality (24, 25, 26). Interestingly, follicular cells have been shown to modulate chromatin configuration and thus global transcriptional activity of the oocyte during its growth (24, 27). Because $Myo10^{-/-}$ full

---

test depending on whether the data followed a Gaussian distribution, P = 0.4459 (n.s, Ctrl full versus $Myo10^{-/-}$ full), P = 0.7355 (n.s, Ctrl full versus Ctrl oo), P > 0.9999 (n.s, Ctrl full versus $Myo10^{-/-}$ oo), P = 0.9905 (n.s, $Myo10^{-/-}$ full versus Ctrl oo), P > 0.9999 (n.s, $Myo10^{-/-}$ full versus $Myo10^{-/-}$ oo), and P > 0.9999 (n.s, Ctrl oo versus $Myo10^{-/-}$ oo). n.s, not significant. **(C)** Scatter plot of the *zona pellucida* texture of control and $Myo10^{-/-}$ full oocytes (in dark gray and green, respectively) and control and $Myo10^{-/-}$ oo oocytes (in light gray and blue, respectively). (n) is the number of oocytes analyzed. Data are the mean ± s.d. with individual data points plotted. Data are from four to 13 independent experiments. Statistical significance of differences was assessed by an ANOVA or a Kruskal–Wallis test depending on whether the data followed a Gaussian distribution, P = 0.0280 (*, Ctrl full versus $Myo10^{-/-}$ full), P = 0.3355 (n.s, Ctrl full versus $Myo10^{-/-}$ oo), P > 0.9999 (n.s, Ctrl full versus $Myo10^{-/-}$ oo), P < 0.0001 (****, $Myo10^{-/-}$ full versus Ctrl oo), P = 0.0083 (**, $Myo10^{-/-}$ full versus $Myo10^{-/-}$ oo), and P = 0.3895 (n.s, Ctrl oo versus $Myo10^{-/-}$ oo). n.s, not significant. **(D)** Brightfield images of a fully grown control (upper panel) and $Myo10^{-/-}$ full oocyte (bottom panel). For each panel, the right image displays the automatic segmentation of the oocyte perivitelline space (perivitelline space, yellow). Scale bar = 10 µm. **(E)** Scatter plot of the coefficient of variation of the perivitelline space thickness, as represented by the schemes on the left. Control and $Myo10^{-/-}$ full oocytes are in dark gray and green, respectively. Control and $Myo10^{-/-}$ oo oocytes are in light gray and blue, respectively. (n) is the number of oocytes analyzed. Data are the mean ± s.d. with individual data points plotted. Data are from four to 13 independent experiments. Statistical significance of differences was assessed by an ANOVA or a Kruskal–Wallis test depending on whether the data followed a Gaussian distribution, P < 0.0001 (****, Ctrl full versus $Myo10^{-/-}$ full), P > 0.9999 (n.s, Ctrl full versus Ctrl oo), P = 0.3139 (n.s, Ctrl full versus $Myo10^{-/-}$ oo), P < 0.0001 (****, $Myo10^{-/-}$ full versus Ctrl oo), P = 0.0103 (*, $Myo10^{-/-}$ full versus $Myo10^{-/-}$ oo), and P = 0.0811 (n.s, Ctrl oo versus $Myo10^{-/-}$ oo). n.s, not significant. **(F)** Images of the flattened *zona pellucida* obtained from oocyte segmentation. The top panel shows a control *zona pellucida*, and the lower one, a $Myo10^{-/-}$ full *zona pellucida*. For each panel, the bottom image is a magnification of the brightfield image of the *zona pellucida* at the top. Scale bar = 10 µm. Below each flattened *zona pellucida* are examples of the *zona pellucida* from control and $Myo10^{-/-}$ full oocytes. Brightfield images are on the left panels, SiR-actin (white)–labeled TZPs are on the middle panels, and merge images (SiR-actin in magenta) are on the right panels. Scale bar = 9 µm. **(G)** Scatter plot of the presence of vertical TZP-like structures in the *zona pellucida* (ZP), as represented by the schemes on the left. Control and $Myo10^{-/-}$ full oocytes are in dark gray and green, respectively. Control and $Myo10^{-/-}$ oo oocytes are in light gray and blue, respectively. (n) is the number of oocytes analyzed. Data are the mean ± s.d. with individual data points plotted. Data are from four to 13 independent experiments. Statistical significance of differences was assessed by an ANOVA or a Kruskal–Wallis test depending on whether the data followed a Gaussian distribution, P < 0.0001 (****, Ctrl full versus $Myo10^{-/-}$ full), P = 0.7183 (n.s, Ctrl full versus Ctrl oo), P = 0.7291 (n.s, Ctrl full versus $Myo10^{-/-}$ oo), P < 0.0001 (****, $Myo10^{-/-}$ full versus Ctrl oo), P < 0.0001 (****, $Myo10^{-/-}$ full versus $Myo10^{-/-}$ oo), and P = 0.9999 (n.s, Ctrl oo versus $Myo10^{-/-}$ oo). n.s, not significant.

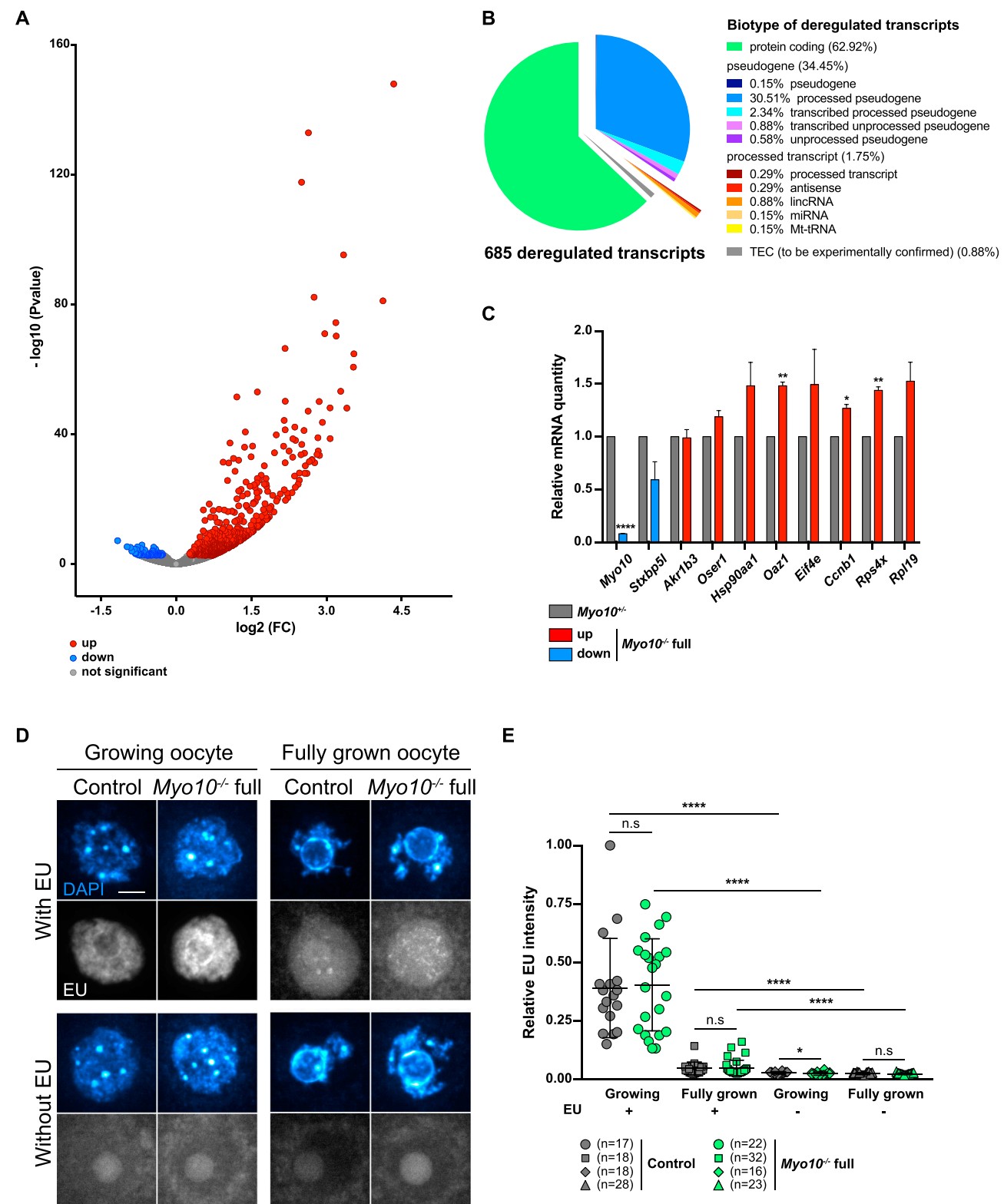

**Figure 3. Gene expression is modified in TZP-deprived oocytes.**
**(A)** Volcano plot of differential gene expression between fully grown *Myo10$^{+/-}$* and *Myo10$^{-/-}$* full oocytes. Differential transcriptomic analysis was performed by RNA-Seq. Up-regulated transcripts are shown in red (up, 605 transcripts); down-regulated transcripts, in blue (down, 80 transcripts, including *Myo10*); and non-deregulated transcripts, in gray (not significant, 15,702 transcripts). Significance was set at *P*-adj < 0.05. *Myo10$^{-/-}$* full oocytes were collected from two *Myo10$^{flox/flox}$; Cre$^+$* mice and *Myo10$^{+/-}$* oocytes from two *Myo10$^{wt/flox}$; Cre$^+$* mice. For each condition, three biological replicates (each containing 10 oocytes) and three technical replicates were

oocytes lack follicular cell contacts and potentially benefit less from exogenous transcriptional regulation, we tested whether their transcriptome differed from that of control oocytes.

By performing differential transcriptomic analysis by RNA-Seq between $Myo10^{+/-}$ and $Myo10^{-/-}$ full oocytes at the end of the growth phase (23), we found 685 genes significantly ($P$-adj < 0.05) misregulated in $Myo10^{-/-}$ full oocytes (Fig 3A and Table S1), most of which were up-regulated (605 of 685 representing 88.3% of deregulated transcripts). Most of the deregulated genes were protein-coding genes (Fig 3B), 10 of which were validated by RT–qPCR (Fig 3C). Among them, $Myo10$ mRNA was confirmed to be down-regulated in $Myo10^{-/-}$ full oocytes, validating both our RNA-Seq analysis and our genetic deletion tool (Fig 3C). Because most of the deregulated transcripts in $Myo10^{-/-}$ full oocytes were up-regulated, this could mean that the reduction in TZP density had resulted in either a higher transcription rate or increased stability of these transcripts.

We thus tested whether this deregulation reflected a change in the global transcriptional rate of TZP-deprived oocytes (23). Fully grown and smaller growing $Myo10^{-/-}$ full oocytes were incubated with the uridine analog 5-ethynyl uridine (EU) to label RNA synthesis (Fig 3D and E). Oocytes were also incubated without EU to serve as a negative control. To measure global RNA synthesis, the EU signal intensity was normalized to the DAPI signal intensity. Interestingly, global RNA synthesis was not different in growing and fully grown $Myo10^{-/-}$ full oocytes compared with control oocytes (Fig 3E).

In summary, we demonstrate here that the lack of somatic contacts caused by decreased TZP density leads to a deregulation of oocyte gene expression at the end of its growth.

### TZP-deprived oocytes are prone to stop development in meiosis I despite correct spindle morphogenesis and chromosome alignment

We showed above that reduction in somatic contacts modulates the oocyte transcriptome. Importantly, oocytes depend on the transcripts synthesized during growth to develop until fertilization (25, 26). We thus tested whether TZP-deprived oocytes successfully underwent the developmental step after growth, namely, meiotic divisions. After nuclear envelope breakdown (NEBD), the oocyte experiences an asymmetric division in size, leading to a large cell, the oocyte, and a tiny cell, the polar body. This size difference is essential to keep the nutrients stored during growth in the prospective fertilized oocyte for embryogenesis (28). Importantly, we observed that $Myo10^{-/-}$ full oocytes deprived of TZPs showed an increased frequency of developmental arrest at meiosis I, without extruding a polar body (Video 4 and Fig 4A and B; 38% of TZP-deprived oocytes do not extrude a polar body, compared with 15% in the controls), consistent with our morphological characterization using Oocytor (Fig 2C). We previously showed that MYO10 expressed in oocytes is dispensable for meiotic divisions and female fertility (16), suggesting that the meiotic arrest observed in a subset of $Myo10^{-/-}$ full oocytes may be a consequence of decreased follicular cell contacts before meiosis resumption. We also observed that the subpopulation of $Myo10^{-/-}$ full oocytes that extruded a polar body did so with a timing similar to controls (Video 4 and Fig 4A and C), indicating that the complete loss of MYO10 and subsequent TZP depletion did not delay polar body extrusion in oocytes, which could achieve it. Therefore, oocytes deprived of TZPs have a lower developmental potential than oocytes that retain a canonical density of follicular cell contacts, reinforcing the importance of the TZP-mediated dialogue between the oocyte and its niche for a fully efficient acquisition of oocyte competency.

To gain insight into the origin of the meiotic arrest observed in some TZP-deprived oocytes, we first investigated whether the meiotic spindles were correctly formed in $Myo10^{-/-}$ full oocytes. After NEBD, microtubules organize into a ball near the chromosomes that gradually bipolarizes to form a barrel-shaped spindle (29, 30, 31). To assess spindle morphogenesis, oocytes were stained with SiR-tubulin (SiR-tub) to label microtubules and followed throughout meiotic maturation. We observed that in both $Myo10^{-/-}$ full oocytes that extruded a polar body and those that did not, meiosis I spindles bipolarized properly and within similar timings as in controls (Video 5 and Fig 5A–D). To further evaluate spindle

performed. **(B)** Pie chart of deregulated transcript biotypes from the RNA-Seq analysis. lincRNA: long intergenic non-coding RNAs; miRNA: microRNA precursors; Mt-tRNA: transfer RNA located in the mitochondrial genome; and TEC: to be experimentally confirmed. **(C)** Bar graph of the relative mRNA quantity in fully grown $Myo10^{-/-}$ full oocytes (colored bars) normalized to fully grown $Myo10^{+/-}$ oocytes (gray bars) performed by RT–qPCR. **(A)** Selected mRNAs were chosen based on their biological relevance and their deregulatory strength ($\log_2$[FC] and $P$-adj) in the RNA-Seq analysis described in (A). mRNAs indicated as up- or down-regulated by RNA-Seq analysis are shown in red and blue, respectively. $Myo10^{-/-}$ full oocytes were collected from $Myo10^{flox/flox}$; $Cre^-$ mice and $Myo10^{+/-}$ oocytes from $Myo10^{wt/flox}$; $Cre^+$ mice. For each condition, two biological replicates were performed (one containing 30 oocytes and the other containing 24 oocytes) from two different mice each time and two technical replicates were performed. The SEM is shown. For each mRNA, the mean of $Myo10^{+/-}$ oocytes was normalized to that of $Myo10^{+/-}$ oocytes. The mean and the SEM for $Myo10^{-/-}$ full oocytes = 0.08 ± 0.004 ($Myo10$); 0.59 ± 0.171 ($Stxbp5l$); 0.99 ± 0.079 ($Akr1b3$); 1.19 ± 0.059 ($Oser1$); 1.48 ± 0.222 ($Hsp90aa1$); 1.48 ± 0.034 ($Oaz1$); 1.49 ± 0.333 ($Eif4e$); 1.27 ± 0.037 ($Ccnb1$); 1.44 ± 0.034 ($Rps4x$); and 1.53 ± 0.179 ($Rpl19$). Statistical significance of differences was assessed by two-tailed unpaired $t$ tests, $P < 0.0001$ ($Myo10$), $P = 0.1394$ ($Stxbp5l$), $P = 0.9001$ ($Akr1b3$), $P = 0.0864$ ($Oser1$), $P = 0.1612$ ($Hsp90aa1$), $P = 0.0050$ ($Oaz1$), $P = 0.2765$ ($Eif4e$), $P = 0.0180$ ($Ccnb1$), $P = 0.0060$ ($Rps4x$), and $P = 0.0987$ ($Rpl19$). **(D)** Sum of Z projection images of oocytes stained with DAPI to label DNA (blue, top images of each panel) and incubated or not with 5-ethynyl uridine to label global RNA transcription (EU, bottom images of each panel). For each panel, control oocytes are on the left, and $Myo10^{-/-}$ full oocytes, on the right. Growing and fully grown oocytes are on the left and right panels, respectively. Oocytes incubated with or without EU are on the upper and lower panels, respectively. For EU staining, contrast adjustment is similar between all conditions except for growing oocytes incubated with EU for which the signal intensity is too high to be shown with the same adjustment as the others. Scale bar = 10 $\mu$m. **(E)** Scatter plot of the relative EU signal intensity normalized to the DAPI signal intensity in control (dark gray) and $Myo10^{-/-}$ full (green) growing or fully grown oocytes incubated or not with EU. (n) is the number of oocytes analyzed. Data are the mean ± s.d. with individual data points plotted. Data are from three to five independent experiments. Statistical significance of differences was assessed by two-tailed Mann–Whitney's tests, $P = 0.7688$ (growing $Myo10^{-/-}$ full oocytes with EU compared to growing control oocytes with EU), $P = 0.7564$ (fully grown $Myo10^{-/-}$ full oocytes with EU compared to fully grown control oocytes with EU), $P = 0.0326$ (growing $Myo10^{-/-}$ full oocytes without EU compared to growing control oocytes without EU), $P < 0.0001$ (growing control oocytes with EU compared to growing control oocytes without EU), $P < 0.0001$ (growing $Myo10^{-/-}$ full oocytes with EU compared to growing $Myo10^{-/-}$ full oocytes without EU), $P < 0.0001$ (fully grown control oocytes with EU compared to fully grown control oocytes without EU), and $P < 0.0001$ (fully grown $Myo10^{-/-}$ full oocytes with EU compared to fully grown $Myo10^{-/-}$ full oocytes without EU). Statistical significance of differences was assessed by a two-tailed unpaired $t$ test to compare fully grown $Myo10^{-/-}$ full oocytes without EU to fully grown control oocytes without EU, $P = 0.2647$. n.s, not significant.

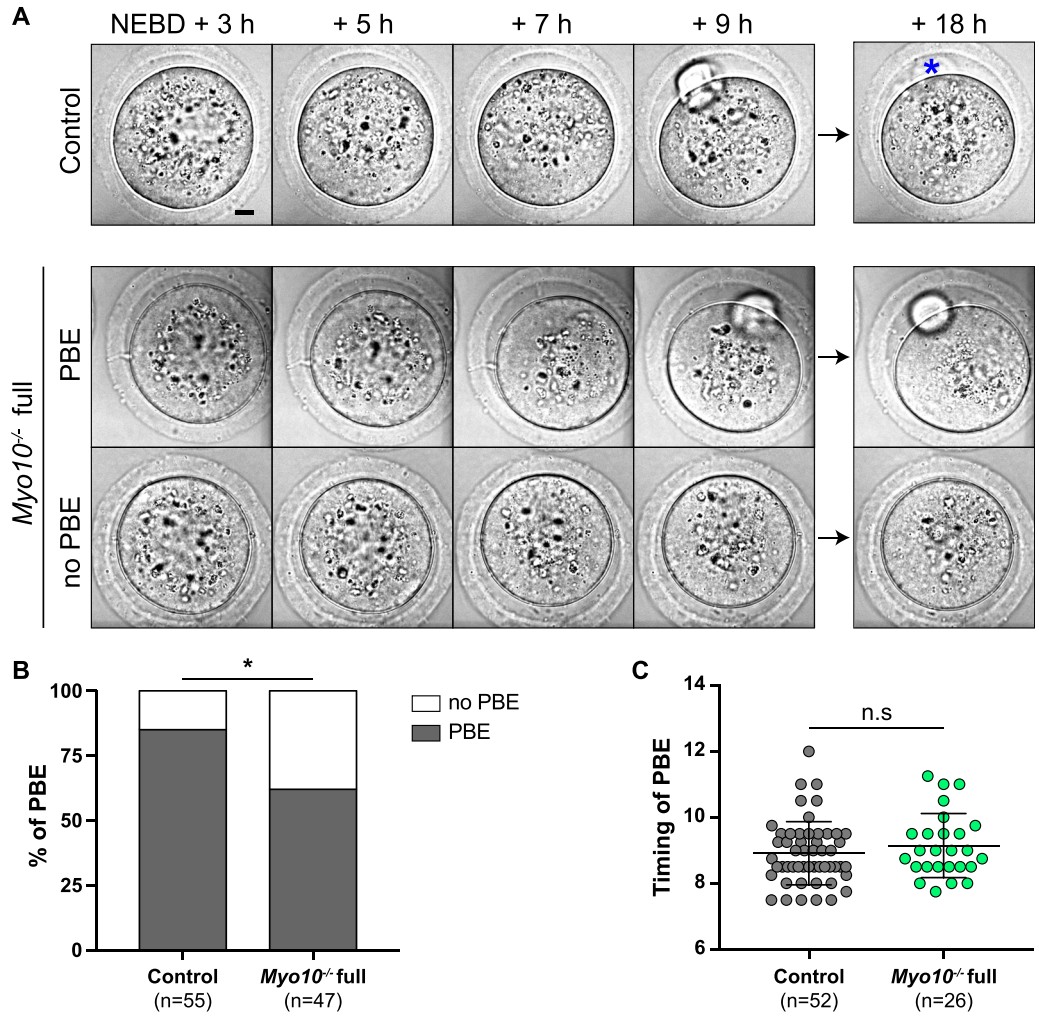

**Figure 4. TZP-deprived oocytes tend to arrest in meiosis I.**
**(A)** Brightfield images from spinning disk videos from 3 h to 18 h after NEBD. The top images show a control oocyte; the middle ones, a $Myo10^{-/-}$ full oocyte that extruded a polar body (PBE); and the bottom ones, a $Myo10^{-/-}$ full oocyte that did not extrude a polar body (no PBE). The blue asterisk indicates a lysed polar body. Scale bar = 10 $\mu$m. **(B)** Stacked bars of first polar body extrusion rate as a percentage of control and $Myo10^{-/-}$ full oocytes. The gray bars represent the percentage of oocytes that extruded a polar body (PBE), and the white bars, the percentage of oocytes that did not extrude a polar body (no PBE). (n) is the number of oocytes analyzed. Data are from three independent experiments. Statistical significance of differences was assessed by a two-sided Fisher exact test, $P = 0.0113$. **(C)** Scatter plot of the timing in hours after NEBD of first polar body extrusion of control (dark gray) and $Myo10^{-/-}$ (green) full oocytes. (n) is the number of oocytes analyzed. Data are the mean ± s.d. with individual data points plotted. Data are from six independent experiments. Statistical significance of differences was assessed by a two-tailed Mann–Whitney test, $P = 0.3194$. n.s, not significant.

morphogenesis in $Myo10^{-/-}$ full oocytes, we measured spindle interpolar length and central width 7 h after NEBD once spindle morphogenesis is complete and showed that $Myo10^{-/-}$ full spindles were not significantly different from control spindles (Fig 5B and D). Importantly, we observed that $Myo10^{-/-}$ full oocytes that did not extrude a polar body arrested development in metaphase I, without experiencing anaphase (Video 5 and Fig 5A).

In addition to assessing spindle formation, we also examined meiosis I spindle positioning in $Myo10^{-/-}$ full oocytes. During meiosis I, the spindle first assembles in the center of the oocyte and then migrates to the cell cortex by a process involving F-actin, resulting in an asymmetric division in size (32, 33, 34, 35). We monitored the positioning of SiR-tub–labeled spindles throughout meiosis I and analyzed their localization 30 min before anaphase

for oocytes that extruded a polar body or up to 12 h after NEBD for those that did not. The percentage of $Myo10^{-/-}$ full oocytes with an off-centered spindle was not different from control oocytes (Video 5 and Fig 5A and E). In agreement with unaltered spindle positioning, we did not observe a defect in the organization of the F-actin cytoplasmic network (including the actin cage) mediating spindle migration in fixed $Myo10^{-/-}$ full oocytes labeled with phalloidin (Fig S5A–C). These results refine our understanding of the meiotic arrest of some $Myo10^{-/-}$ full oocytes by showing that they arrest in metaphase I with a normally assembled and positioned spindle.

Because severe chromosome misalignment could prevent anaphase onset by sustaining the spindle assembly checkpoint in mouse oocytes (36), we assessed chromosome alignment in

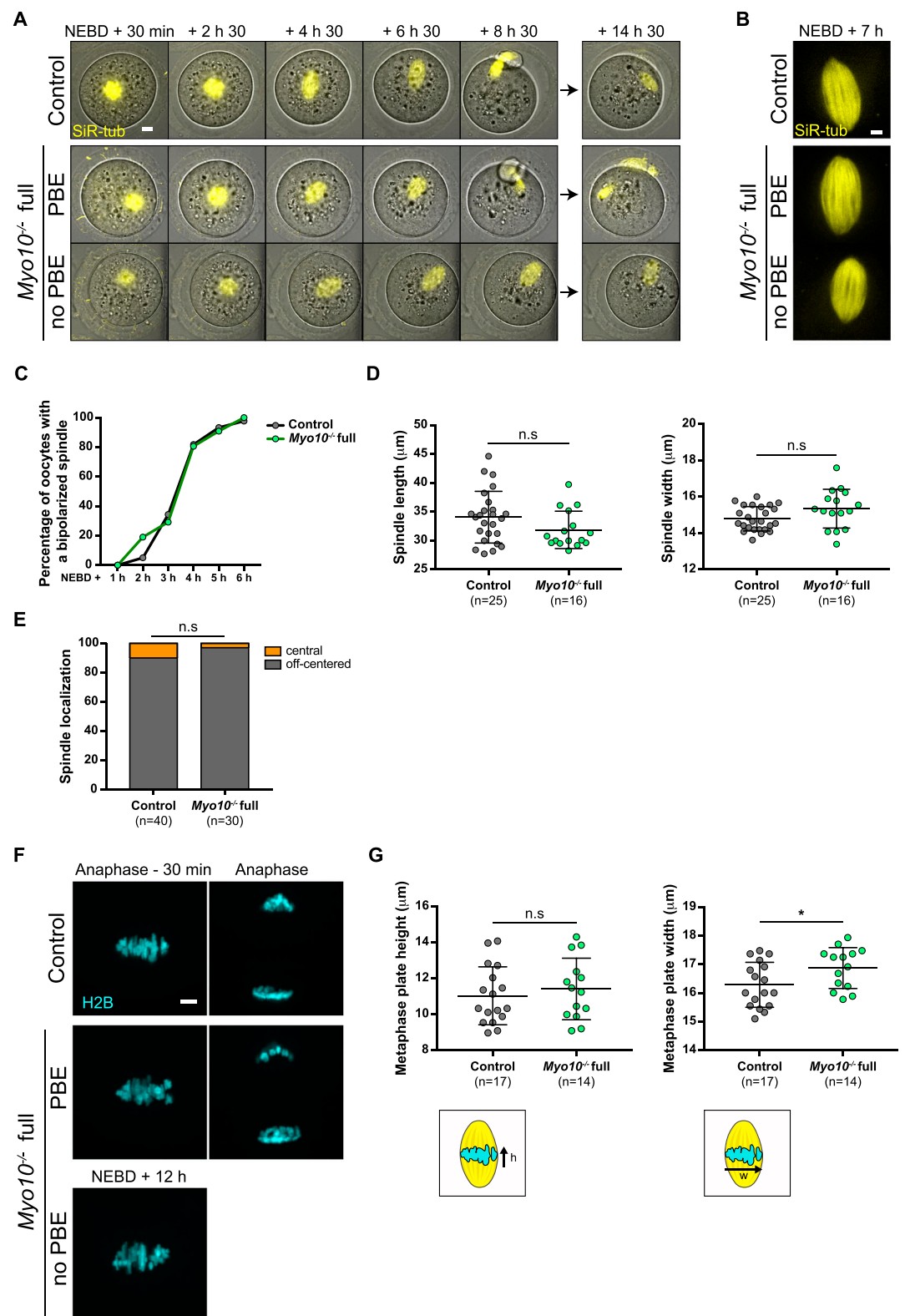

**Figure 5. TZP-deprived oocytes have correctly formed and positioned spindles and aligned chromosomes.**
**(A)** Maximum-intensity Z projection images of oocytes stained with SiR-tubulin to label microtubules (SiR-tub, yellow) merged with corresponding brightfield images. Images are extracted from spinning disk videos from 30 min to 14 h 30 min after NEBD. The top images show a control oocyte; the middle ones, a Myo10$^{-/-}$ full oocyte that extruded a polar body (PBE); and the bottom ones, a Myo10$^{-/-}$ full oocyte that did not extrude a polar body (no PBE). Scale bar = 10 μm. **(B)** Maximum-intensity Z projection images of oocytes stained with SiR-tub (yellow) 7 h after NEBD. The top image shows the spindle of a control oocyte; the middle one, the spindle of a Myo10$^{-/-}$ full oocyte

*Myo10^{-/-}* full oocytes deprived of TZPs. Oocytes were injected with H2B-GFP to label chromosomes, and chromosome alignment was monitored during meiosis I (Video 6 and Fig 5F and G). Interestingly, metaphase I chromosomes were aligned on the metaphase plate similar to controls in *Myo10^{-/-}* full oocytes, as evidenced by a metaphase plate height not significantly different from controls 30 min before anaphase for oocytes that extruded a polar body or 12 h after NEBD for those that did not (Fig 5G). Therefore, the metaphase I arrest of some *Myo10^{-/-}* full oocytes is not caused by severe chromosome misalignment. Its origin remains to be further investigated.

### TZP-deprived oocytes produce fewer embryos, and some cease development before the blastocyst stage, correlating with subfertility in females

Because TZP-deprived oocytes display oocyte-matrix defects, deregulated gene expression, and a tendency to arrest in meiosis I, we sought to assess their developmental potential after fertilization. For that, we superovulated female littermates (*Myo10^{-/-} full* and control animals), mated them with male controls, and recovered embryos at E0.5, that is, at the zygote stage. First, we recovered more embryos for controls than for *Myo10^{-/-} full* animals for the same number of females (Fig 6A; compare the n, and compare the number of cells recovered from each female in the right images). Second, we recovered more unfertilized oocytes and more dead embryos in the *Myo10^{-/-} full* females compared with the controls (Fig 6A, black and gray bars, respectively, and compare the number of two-cell-stage embryos at E1.5 for each female in the right images). Finally, we saw that among the embryos that survived, some stopped their development before reaching the blastocyst stage (Fig 6B, black bars). All this suggests that the developmental potential of TZP-deprived oocytes is impaired. We therefore evaluated the fertility of *Myo10^{-/-} full* females by mating them with control males. The number of litters per couple, the mean number of pups per litter, and thus the total number of pups per couple were significantly lower for *Myo10^{-/-} full* females than for controls (Fig 6C), showing that *Myo10^{-/-} full* females are subfertile and arguing that TZP-mediated communication is important for the developmental potential of the oocyte and for female fertility.

## Discussion

In this study, we aimed to gain more insight into the contribution of cellular protrusions established by surrounding somatic cells to postnatal oocyte development. By genetically deleting the TZP component MYO10 in all murine cell types, including follicular cells, we greatly reduced the density of oocyte–follicular cell contacts and assessed the resulting oocyte development.

Surprisingly, although the complete loss of MYO10 decreased by fivefold the density of TZPs in fully grown oocytes, *Myo10^{-/-} full* oocytes grew to normal sizes (Fig 2A and B) and were arrested in prophase. Hence, and in contrast to previous studies where the impairment of oocyte–follicular cell communication by deletion of connexin 37 or GDF9 led to, respectively, smaller or larger oocytes than controls (17, 18, 19), reducing TZP density did not prevent oocytes from growing to a canonical size. Interestingly, a small fraction of TZPs remained in *Myo10^{-/-} full* oocytes, and these remaining TZPs were functional allowing exchanges between the oocyte and follicular cells (Fig S2C and D). This implies that a subpopulation of TZPs may be MYO10-independent and potentially explains how oocytes managed to grow to a canonical size and maintained the meiotic arrest in prophase despite global reduced intercellular communication.

Even though TZP-deprived oocytes grew to normal sizes, we identified morphological differences, mostly related to the integrity of the *zona pellucida*. The *zona pellucida*, which is the extracellular matrix surrounding the oocyte, is composed of proteins secreted by the oocyte throughout growth and assembled in cross-linked filaments (37, 38). Using our machine-learning algorithm to phenotype oocytes, we found that the *zona pellucida* was more homogeneously detached in TZP-deprived oocytes. It is likely that reduction in TZP density, and consequently the decrease in adherens junctions between the oocyte and follicular cells (9, 11), led to this apparent looser oocyte-matrix adhesion. Furthermore, our machine-learning algorithm detected TZP-like structures more readily in *Myo10^{-/-} full* oocytes than in controls, despite their decrease in TZP density. These structures always contain actin (Fig 2F). Thus, there are less TZPs in *Myo10* full knockouts (Fig 1E), but most of these remaining TZPs are visible by transmitted light. This could reflect an alteration in the structure of the remaining TZPs, or alternatively a difference in the structure of the *Myo10^{-/-} full zona*

---

that extruded a polar body (PBE); and the bottom one, the spindle of a *Myo10^{-/-}* full oocyte that did not extrude a polar body (no PBE). Scale bar = 5 µm. **(C)** Graph of the percentage of control (dark gray) and *Myo10^{-/-}* (green) full oocytes with a bipolarized spindle as a function of time after NEBD. Data are from four independent experiments. For each time point, statistical significance of differences was assessed by a two-sided Fisher exact test, $P > 0.9999$ for 1 h after NEBD, $P = 0.3433$ for 2 h after NEBD, $P = 0.7765$ for 3 h after NEBD, $P > 0.9999$ for 4 h after NEBD, $P = 0.6937$ for 5 h after NEBD, and $P > 0.9999$ for 6 h after NEBD. **(D)** Scatter plots of spindle interpolar length (left) and central width (right) 7 h after NEBD in control (dark gray) and *Myo10^{-/-}* (green) full oocytes. (n) is the number of oocytes analyzed. Data are the mean ± s.d. with individual data points plotted. Data are from four independent experiments. Statistical significance of differences was assessed by a two-tailed unpaired *t* test, $P = 0.0989$ for the length, and by a two-tailed unpaired *t* test with Welch's correction, $P = 0.0753$ for the width. n.s, not significant. **(E)** Stacked bars of spindle cell localization 30 min before anaphase for oocytes that extruded a polar body or 12 h after NEBD for those that did not, as a percentage of control and *Myo10^{-/-}* full oocytes. Gray bars represent the percentage of oocytes with an off-centered spindle, and orange bars, the percentage of oocytes with a central spindle. (n) is the number of oocytes analyzed. Data are from four independent experiments. Statistical significance of differences was assessed by a two-sided Fisher exact test, $P = 0.3832$. n.s, not significant. **(F)** Maximum-intensity Z projection images from spinning disk videos of oocytes injected with H2B-GFP to label chromosomes (H2B, cyan) in a control oocyte (top images), a *Myo10^{-/-}* full oocyte that extruded a polar body (PBE, middle images), and a *Myo10^{-/-}* full oocyte that did not extrude a polar body (no PBE, bottom image). Left images show chromosomes 30 min before anaphase for polar body–extruding oocytes or 12 h after NEBD for the *Myo10^{-/-}* full oocyte that did not extrude a polar body. Right images show chromosomes after anaphase. Scale bar = 5 µm. **(G)** Scatter plots of metaphase plate height (left) and width (right), as shown in the lower schemes, 30 min before anaphase for oocytes that extruded a polar body or 12 h after NEBD for those that did not. Control and *Myo10^{-/-}* full oocytes are in dark gray and green, respectively. (n) is the number of oocytes analyzed. Data are the mean ± s.d. with individual data points plotted. Data are from three independent experiments. Statistical significance of differences was assessed by two-tailed unpaired *t* tests. $P = 0.5331$ for the height and $P = 0.0395$ for the width. n.s, not significant.

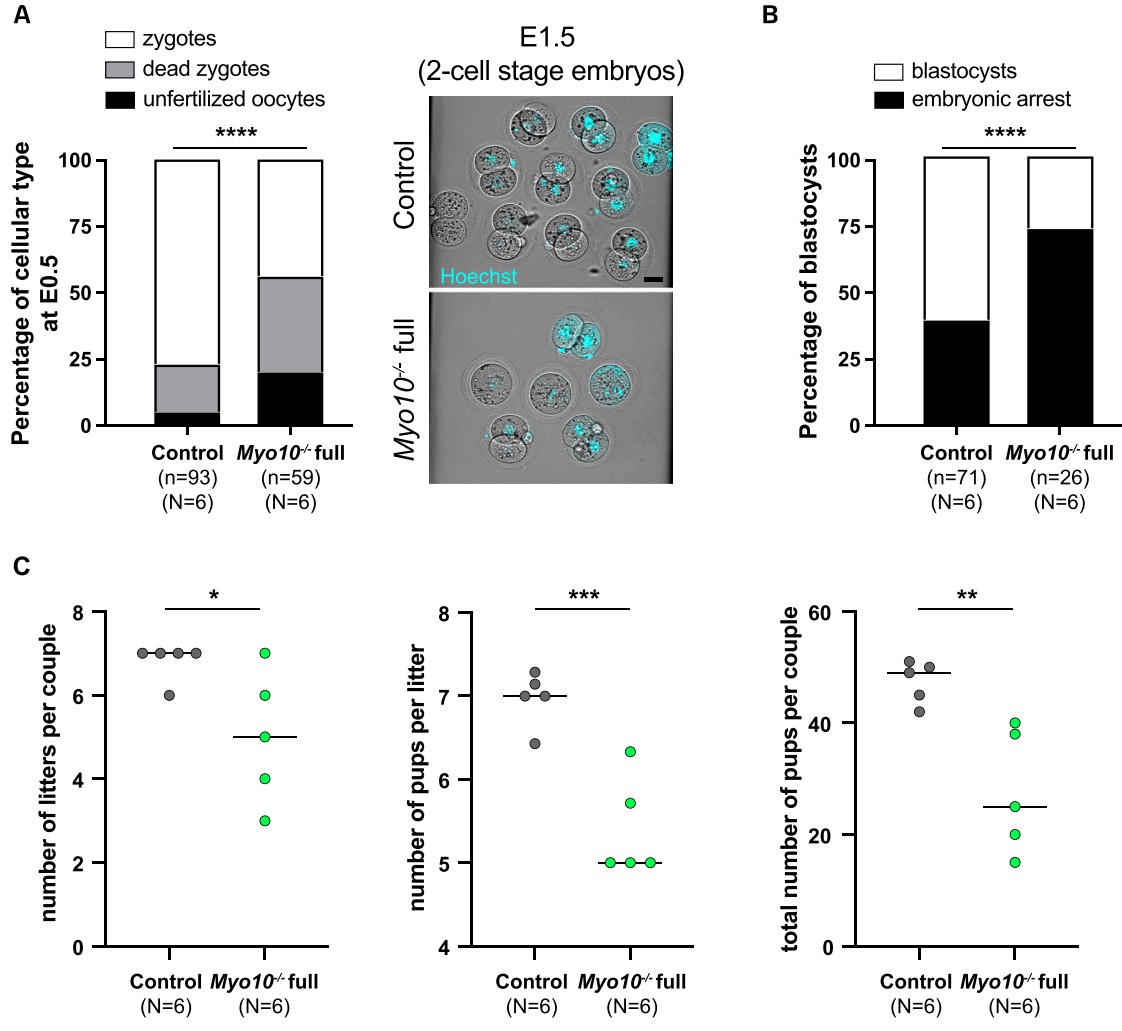

**Figure 6. Developmental potential of TZP-deprived oocytes is altered, correlating with subfertility in females.**
**(A)** On the left, stacked bars showing the percentage of unfertilized oocytes (black bars), dead zygotes (gray bars), and alive zygotes (white bars) recovered at E0.5. (n) is the number of cells recovered, and (N) is the number of successful matings analyzed. Data are from three independent experiments. Statistical significance of differences was assessed with a chi-squared test, $P < 0.0001$. On the right, cells recovered at E0.5 and cultured to E1.5 from a control female (upper panel) and a $Myo10^{-/-}$ full female (lower panel), stained with Hoechst (blue) to visualize DNA. Scale bar = 40 µm. **(B)** Stacked bars showing the percentage of embryonic arrest (black bars) and blastocysts (white bars). (n) is the number of cells recovered, and (N) is the number of successful matings analyzed. Data are from three independent experiments. Statistical significance of differences was assessed by a two-sided Fisher exact test, $P < 0.0001$. **(C)** Scatter plots of the number of litters per couple, number of pups per litter, and total number of pups per couple from $Myo10^{wt/wt}$; $Cre^+$ female mice (control, gray) and $Myo10^{flox/flox}$; $Cre^+$ female mice (carrying $Myo10^{-/-}$ full oocytes, green). Six independent matings (N) were set for six months each with $Myo10^{wt/wt}$; $Cre^-$ males. Statistical significance of differences was assessed with two-tailed unpaired t tests, $P = 0.04$ (for the number of litters per couple), $P = 0.0009$ (for the number of pups per litter), and $P = 0.0052$ (for the total number of pups per couple).

*pellucida* that would make the remaining protrusions crossing it more apparent. All these features taken together may reflect impaired assembly or maintenance of the *zona pellucida* in oocytes with reduced cell–cell contacts, because the *zona pellucida* is a viscoelastic extracellular matrix whose properties could change as a function of the cross-linking generated by the number of TZPs passing through it (37, 38). In support of this, we showed using AFM that the reduced number of TZPs passing through it affects the structure, and thus the mechanical properties of the *zona pellucida*, decreasing its elasticity (Fig S4H). In addition, we occasionally observed some follicular-like cells located ectopically within the perivitelline space of TZP-deprived oocytes, a phenotype not observed in control oocytes (Fig S4C; see red arrow). The same ectopic

localization was observed in GDF9-deficient mice, which also displayed oocytes with reduced TZP density, and such abnormal localization of follicular cells was correlated with low oocyte viability (19). Interestingly, the presence of follicular-like cells inside the perivitelline space has been observed in mice lacking ZP1 (39), the secreted *zona pellucida* protein that mediates the cross-linking of the matrix filaments (37, 38). By modulating matrix filament cross-linking, TZPs may therefore be involved in preserving *zona pellucida* integrity, which is important for fertilization (37, 38). Interestingly, we recovered more unfertilized oocytes from $Myo10^{-/-}$ *full* females compared with controls after mating, which suggests a potential effect of the observed alteration of the *zona pellucida* on fertilization (Fig 6A).

Importantly, TZP-deprived oocytes have lower developmental competence than control oocytes, as they more frequently arrest at metaphase of the first meiotic division, and produce fewer embryos with lower viability, impairing female fertility. Consistent with our results, a decrease in oocyte developmental potential correlated with reduced TZP density was also observed in oocytes lacking the focal adhesion kinase/proline-rich tyrosine kinase 2 (40) and oocytes lacking radixin (41). In $Myo10^{-/-}$ full oocytes, the basis of the meiotic arrest observed in a subpopulation of oocytes remains to be deciphered. This developmental arrest may be the readout of the altered oocyte transcriptome because we revealed by RNA-Seq analysis that gene expression before meiosis resumption is modified in TZP-deprived oocytes. As oocytes undergo meiotic divisions relying on transcripts synthesized and stored during growth (25, 26), an impairment in this storage could compromise oocyte quality. Interestingly, some deregulated transcripts in TZP-deprived oocytes, including the most deregulated ones, are involved in the oxidative stress response, the metabolism of polyamine, and cell cycle transitions (Oser1, Akr1b3, Oaz1, Psma5, and Cyclin B1; Fig 3C and Table S1), pathways that when deregulated are known to affect subsequent meiotic maturation and early embryogenesis (42, 43, 44, 45, 46, 47).

In conclusion, we have shown that cytoplasmic protrusions of surrounding somatic cells enhance the developmental potential of the oocyte. Consistent with other studies, we suggest that TZPs contribute to zona pellucida integrity and oocyte-matrix adhesion—directly as an intercellular contact structure or indirectly through metabolite exchange optimizing oocyte quality and the establishment of a robust zona pellucida—thus playing a role in the structural integrity of the ovarian follicle. In addition, we propose a novel role of TZPs as modulators of the synthesis/stability of specific oocyte transcripts at the end of growth that are required for successful meiotic divisions and early embryonic development. Interestingly, as the density of TZPs decreases with maternal age (7), investigating their contribution to oocyte quality could be of great benefit to further unraveling female fertility, which decreases with maternal age. This may be of particular interest to increase the efficiency of assisted reproductive technologies. More broadly, our results may be valuable for the comprehension of other cellular models relying on distant cell–cell contact communication to function (48). It may notably provide clues as to how cellular development is modulated by exogenous regulation of gene expression through direct intercellular dialogue.

## Materials and Methods

### Mouse strain generation and genotyping

To generate $Myo10^{wt/flox}$ mice, two loxP sites flanking Myo10 exons 23–25 and an FRT-flanked neomycin selection cassette located between Myo10 exons 22 and 23 were inserted in the genome of C57BL/6 mice by genOway. Mice were then mated with Zp3-Cre mice already housed in the laboratory.

To obtain Myo10 full knockout mice ($Myo10^{-/-}$ full), the neomycin selection cassette was retained to allow its promoter to interfere with Myo10 expression independently of Cre expression (49)

(Fig S1A). We used the following primers to detect the presence of the neomycin selection cassette at the Myo10 locus: 5′-ACA GCC CAT ATC ACT GTC TAG AGA CCC ATT-3′, 5′-ATC GCC TTC TAT CGC CTT GAC G-3′, and 5′-GAG GAT CCA GAC TTG GAC CCG GTC-3′.

Mice conditionally deleted for Myo10 in oocytes ($Myo10^{-/-}$ oo mice) were previously generated from the above strain by removing the neomycin selection cassette after mating with FLPo mice (16) (Fig S1A). To detect Cre-mediated excision at the Myo10 locus after removing the neomycin selection cassette, the following PCR primers were used: for the Cre, 5′-GCG GTC TGG CAG TAA AAA CTA TC-3′ and 5′-GTG AAA CAG CAT TGC TGT CAC TT-3′; and for the Myo10, 5′-ACC CCA GTA CTT GTT CAT ACA TCC TAT ATC CTA CA-3′, 5′-GAC TAC ACC ATT CTG AAT GTG CCT GAT CTC-3′, and 5′-GAG TAT CTG CCA TCT TGT CCC TAA AGG TGG-3′.

For the Myo10 full knockout strain, we used $Myo10^{wt/wt}$; $Cre^+$, $Myo10^{wt/flox}$; $Cre^+$, $Myo10^{wt/\Delta}$; $Cre^+$, and $Myo10^{wt/flox}$; $Cre^-$ as control mice, and $Myo10^{flox/flox}$; $Cre^+$, $Myo10^{flox/\Delta}$; $Cre^+$, and $Myo10^{flox/flox}$; $Cre^-$ as $Myo10^{-/-}$ full mice. For the Myo10 oocyte knockout strain, we used $Myo10^{wt/flox}$; $Cre^+$, $Myo10^{wt/flox}$; $Cre^-$, and $Myo10^{flox/flox}$; $Cre^-$ as control mice, and $Myo10^{flox/flox}$; $Cre^+$ as $Myo10^{-/-}$ oo mice.

### Immunoblotting

Immunoblotting was performed as in Reference 14 on extracts from $Myo10^{wt/flox}$; $Cre^+$, $Myo10^{flox/flox}$; $Cre^+$, and $Myo10^{flox/flox}$; $Cre^-$ mice from the Myo10 full knockout strain. Whole liver, whole spleen, and one kidney per mouse were extracted, weighed, and immediately added to 1× Laemmli sample buffer (50 mM Tris, pH 7.4, 2% SDS, 6% glycerol, 0.1 M DTT, and 0.25% bromophenol blue) at 1 ml/20 mg wet tissue weight and boiled for 15 min. Samples were crushed with a glass potter to dissociate the tissue, and the lysates were centrifuged at 20,000$g$ for 30 min at room temperature. The pellets were discarded, and the supernatants were stored at −80°C. Final loading volumes were adjusted for immunoblot based on quantification of the actin band from preliminary tests so that the same protein amount is deposited in all the wells. Samples were boiled for 1 min and subjected to SDS–PAGE using a 4–15% Mini-PROTEAN TGX Stain-Free Protein Gel (Ref. 4568085; Bio-Rad). After SDS–PAGE, lysates were transferred to Immobilon-P PVDF Membrane (Ref. IPVH00010; Sigma-Aldrich) at room temperature in transfer buffer (25 mM Tris, 192 mM glycine, 0.3% SDS, and 20% methanol) using Trans-Blot SD Semi-Dry Transfer Cell (Bio-Rad) set at 100 mA constant power with a volt limit of 25 V for 45 min. Before transfer, the PVDF membrane was wetted in methanol, rinsed in distilled water, and equilibrated for 10 min in transfer buffer. After the transfer, the membrane was washed in 50 mM Tris, 150 mM NaCl, and 0.05% Tween-20 (TBST thereafter) and blocked for 1 h in TBST containing 5% dried milk powder. The membrane was washed in TBST and incubated for 1 h in TBST containing 4% dried milk and rabbit anti-MYO10 (1:300, Ref. HPA024223; Sigma-Aldrich) as primary antibodies. The membrane was washed in TBST, blocked for 1 h in TBST containing 5% dried milk powder, rinsed in TBST, and incubated for 45 min in TBST containing 4% dried milk and donkey anti-rabbit IgG HRP-linked (1:10,000, Ref. NA934; Amersham) as secondary antibodies. The membrane was washed in TBST and incubated for 5 min in SuperSignal West Femto Maximum Sensitivity Substrate (Ref. 34095; Thermo Fisher Scientific). Protein expression

was revealed using a chemiluminescence image analyzer (FUSION FX; Vilber, 20-s illumination).

The membrane was then rinsed in TBST and stripped for 10 min in Restore Western Blot Stripping Buffer (Ref. 21059; Life Technologies). The membrane was washed in TBST and incubated for 45 min in TBST containing 4% dried milk and HRP mouse anti-beta-actin AC-15 (1:25,000, Ref. ab49900; Abcam). The membrane was washed in TBST and incubated for 1 min in ECL Western Blotting Substrate (Ref. 32109; Thermo Fisher Scientific). Protein expression was revealed using a chemiluminescence image analyzer (FUSION FX; Vilber, 20-s illumination).

### Whole ovary immunostaining and clearing

Ovaries collected from five control mice (*Myo10^{wt/flox}*; *Cre^+*, between 9 and 12 wk old) and five *Myo10^{-/-}* full mice (*Myo10^{flox/flox}*; *Cre^+*, between 9 and 12 wk old) were fixed in 1X PBS and 4% paraformaldehyde for 4 h with agitation at room temperature, and subsequently washed in 1X PBS where they were stored at 4°C. Bleaching, immunostaining, and clearing were performed as in Reference 20. Ovaries were first dehydrated in increasing concentrations of methanol (1X PBS, methanol 50% and 80%, and 100% methanol twice) for 1 h each time with agitation at room temperature. Ovaries were then bleached in 100% methanol + 6% hydrogen peroxide overnight at 4°C and rehydrated in decreasing concentrations of methanol (100% twice, 1X PBS, and methanol 80% and 50%) for 1 h each time with agitation at room temperature. Ovaries were blocked and permeabilized in 1X PBS, 0.2% gelatin, and 0.5% Triton X-100 (PBSGT thereafter) for 4 d with rotation at room temperature. Antibody incubations were performed in PBSGT + 10X saponin (10 mg/ml) at 37°C with rotation at 70*g* for 1 wk for primary antibodies and overnight for secondary antibodies. Washes between antibody incubations and after secondary antibody incubation were done in PBSGT for 1 d with rotation at room temperature. We used rabbit anti-MYO10 (1:1,000, Ref. HPA024223; Sigma-Aldrich) and rat anti-MECA-32 (1:500, Ref. 550563; BD Pharmingen, data not shown) as primary antibodies and Cy3-conjugated donkey anti-rabbit IgG (1:500, Ref. 711-165-152; Jackson Immuno-Research) and Alexa Fluor 790–conjugated donkey anti-rat IgG (1:250, Ref. 712-655-153; Jackson ImmunoResearch) as secondary antibodies (data not shown). Incubation with TO-PRO-3 (1:100, Ref. T3605; Thermo Fisher Scientific) was performed along with secondary antibody incubation to stain nucleic acid. Ovaries were embedded in 1X TAE and 1.5% agarose, cleared using an adapted 3DISCO clearing protocol with rotation at 12*g*, and protected from light at room temperature. Ovaries were dehydrated in $H_2O$ and 50% tetrahydrofuran (THF anhydrous with 250 ppm butylated hydroxytoluene inhibitor, Ref.186562; Sigma-Aldrich) overnight, and in $H_2O$, 80% THF, and 100% THF twice for 1 h 30 each time. Ovaries were then incubated in dichloromethane (DCM, Ref.270997; Sigma-Aldrich) for 30 min for delipidation and cleared overnight in benzyl ether (DBE, Ref.108014; Sigma-Aldrich).

### Follicle/oocyte collection and culture

Preantral follicles, cumulus–oocyte complexes, and oocytes isolated from follicular cells were collected from 2- to 5-mo-old mice in M2 + BSA medium supplemented with 1 $\mu$M milrinone (50) to maintain oocytes arrested in prophase I. Oocytes were released from follicular cells by repeated pipette aspirations. Meiosis resumption was induced by transferring oocytes to M2 + BSA medium free of milrinone. Culture and live imaging were performed at 37°C under oil. To assess the rate of polar body extrusion, oocytes were cultured overnight in M2 + BSA medium in Heracell 150i $CO_2$ Incubator (Thermo Fisher Scientific) and analyzed the next day for polar body extrusion.

### Embryo recovery and culture

Embryos were isolated from superovulated female mice (*Myo10^{flox/flox}*; *Cre^+* and *Myo10^{flox/flox}*; *Cre^-* as *Myo10^{-/-}* full mice, and *Myo10^{wt/wt}*; *Cre^+* and *Myo10^{wt/wt}*; *Cre^-* as control mice) mated with male mice (*Myo10^{wt/wt}*; *Cre^-*). Superovulation of female mice was induced by intraperitoneal injection of 5 IU pregnant mare's serum gonadotropin (Ceva, Syncro-part), followed by intraperitoneal injection of 5 IU human chorionic gonadotropin (MSD Animal Health, Chorulon) 44–48 h later. Embryos were recovered at E0.5 from plugged females by opening the ampulla followed by a brief treatment with 37°C 0.3 mg/ml hyaluronidase (H4272-30MG; Sigma-Aldrich) and washing in 37°C FHM. When ampulla was not present, embryos were recovered by flushing oviducts with 37°C FHM (MR-122-D; Millipore) using a modified syringe (1400 LL 23; Acufirm). Embryos were handled using an aspirator tube (A5177-5EA; Sigma-Aldrich) equipped with a glass pipette pulled from glass micropipettes (Blaubrand intraMark or Warner Instruments). Embryos were placed in KSOM (MR-107-D; Millipore) supplemented with 0.1% BSA (A3311; Sigma-Aldrich) in 10 ml droplets covered in mineral oil (M8410; Sigma-Aldrich). Embryos were cultured in an incubator under a humidified atmosphere supplemented with 5% $CO_2$ at 37°C for 5 d. Embryos were scored for survival and embryonic stage from E0.5 to E4.5.

### Fluorescent probes for live imaging

Oocytes were incubated for 1 h in M2 + BSA, milrinone medium supplemented with 100 nM SiR-actin (Ref. SC006; Spirochrome) to label F-actin, 5 $\mu$g/ml FM 1–43 (Ref. T35356; Thermo Fisher Scientific) to label membranes, or 100 nM SiR-DNA (Ref. SC007; Spirochrome) to label DNA. Oocytes were incubated in M2 + BSA medium supplemented with 100 nM SiR-tub (Ref. SC006; Spirochrome) at least 1 h before the start of spinning disk videos and throughout video acquisitions to label microtubules. Embryos at E1.5 were incubated for 15 min in M2 + BSA medium supplemented with 5 ng/ml Hoechst (Ref. H3570; Invitrogen) to label DNA.

### In vitro cRNA synthesis and oocyte microinjection

The pRN3–Histone (H2B)–GFP plasmid (51) was linearized with the SfiI restriction enzyme and in vitro–transcribed using mMESSAGE mMACHINE T3 Transcription Kit (Ref. AM1348; Ambion). cRNAs were purified using RNeasy Mini Kit (Ref. 74104; QIAGEN), centrifuged at 4°C, and microinjected into prophase I–arrested oocytes with a FemtoJet microinjector (Eppendorf). Oocytes were maintained in prophase I for 2 h to express cRNAs before meiosis resumption.

## Analysis of cumulus–oocyte coupling

The oocyte of each cumulus–oocyte complex was microinjected with 10% Lucifer Yellow (Ref. L453; Thermo Fisher Scientific) with a FemtoJet microinjector (Eppendorf). The complexes were incubated for 30 min to allow dye transfer to follicular cells and then examined using spinning disk microscopy. To block gap junction transmission, some complexes were incubated in 150 $\mu$M CBX (Ref. C4790; Sigma-Aldrich) for 30 min before, during, and after microinjection. Culture before, during, and after microinjection was performed in M2 + BSA medium supplemented with 1 $\mu$M milrinone and 100 nM SiR-actin.

## AFM measurements

After removing the cumulus cells, oocytes were immobilized on an electronic microscopy hexagonal nickel grid (Ref. DT300H-Ni, $\varnothing$ 3.05 mm, thickness 18 $\mu$m, hole 73 $\mu$m; Gilder) attached to the surface of a petri dish in M2 + BSA medium supplemented with 1 $\mu$M milrinone. AFM-indentation measurements were performed using a Nanowizard IV AFM (JPK Instruments) coupled to a widefield microscope (Zeiss Axio Observer with Hamamatsu sCMOS Flash 4.0 Camera). The contact mode was used. MLCT-C tips (Bruker, with silicon nitride cantilevers) with a 17° side angle, a 5-$\mu$m average height pyramidal tip, and a 20-nm tip radius were used. For all experiments, before measurements, the spring constant and the sensitivity were calibrated using the contact-based method in the petri dish in Milli-Q water. The spring constant was in a range of 0.05–0.015 N/m. Measurements were performed on a squared grid of 5 $\mu$m × 5 $\mu$m at the center of the oocyte. For each oocyte, 16 force–displacement curves were acquired, with a 1 $\mu$m/s approach velocity and a 0.5 nN set point. The measurement takes no longer than 5 min for each oocyte. With JPK Data Processing software, the force–displacement curves were processed by correcting the piezo height with the deflection of the cantilever to get the vertical tip position. Finally, a Sneddon model was fitted on the approach force curves to obtain apparent Young's modulus (52).

## Immunostaining and global transcription detection

Preantral follicles, cumulus–oocyte complexes, and oocytes isolated from follicular cells were immunostained as in Reference 16. Incorporation and detection of 5-ethynyl uridine (EU) into nascent oocyte RNAs were performed similar to Reference 23. For all protocols, follicles, complexes, and oocytes were washed in M2 + PVP medium and adhered to gelatin and polylysine-coated coverslips before fixation (53).

### Phalloidin labeling of TZPs
Preantral follicles and cumulus–oocyte complexes were fixed and permeabilized in 1X PBS, Hepes (100 mM, pH 7), EGTA (50 mM, pH 7), and MgSO$_4$ (10 mM) buffer with 2% Triton X-100 and 0.3% formaldehyde for 30 min at 37°C. Follicles and complexes were then washed in 1X PBS where they were stored overnight or 2 d at 4°C. Follicles and complexes were further permeabilized in 1X PBS and 0.5% Triton X-100 for 10 min and washed in 1X PBS, and 1X PBS and 0.1% Tween-20 (PBSTw thereafter). Blocking and antibody

incubations were performed in PBSTw and 3% BSA at room temperature for 30 min for blocking, 1 h 30 min for primary antibody incubation, and 1 h for secondary antibody incubation. Washes between antibody incubations and after secondary antibody incubation were done in PBSTw at room temperature, and an additional wash in 1X PBS was performed before mounting. For spinning disk and OMX microscopy, follicles and complexes were mounted on slide wells filled with VECTASHIELD Antifade Mounting Medium with or without DAPI (Ref. H-1200 and H-1000; Vector Laboratories). We used rabbit anti-MYO10 (1:200, Ref. HPA024223; Sigma-Aldrich) as primary antibodies and Cy3-conjugated donkey anti-rabbit IgG (1:150, Ref. 711-165-152; Jackson ImmunoResearch) as secondary antibodies. Incubation with Alexa Fluor 488–conjugated phalloidin (10 U/ml, Ref. A12379; Thermo Fisher Scientific) was performed concurrently with secondary antibody incubation to stain F-actin. For STED microscopy, cumulus–oocyte complexes and oocytes freed from follicular cells were mounted on slide wells filled with Abberior Liquid Antifade Mounting Medium (Ref. MM-2009). We used rabbit anti-MYO10 (1:200, Ref. HPA024223; Sigma-Aldrich) as primary antibodies and Abberior STAR 580–conjugated anti-rabbit (1:200, Ref. ST580-1002) as secondary antibodies. Incubation of Abberior STAR RED–conjugated phalloidin (1:200, Ref. STRED-0100) was performed with secondary antibody to stain F-actin.

### Phalloidin labeling of oocyte cytoplasmic F-actin
Oocytes isolated from antral follicles were cleared of the *zona pellucida* in acid Tyrode's solution 1 h–1 h 30 before fixation. Immunostaining was performed using a modified protocol from Reference 33. Oocytes were fixed and permeabilized 6 h after NEBD in 1X PBS, Hepes (100 mM, pH 7), EGTA (50 mM, pH 7), and MgSO$_4$ (10 mM) buffer with 0.2% Triton X-100 and 2% formaldehyde for 30 min at 37°C. Oocytes were then washed in 1X PBS and 0.1% Triton X-100 (PBSTr thereafter) where they were stored overnight at 4°C. The remaining immunostaining protocol is similar to that for phalloidin TZP labeling for spinning disk and OMX microscopy, except that oocytes were directly blocked the day after fixation and blocking and antibody incubations were performed in PBSTr and 3% BSA. Oocytes were mounted on slide wells filled with VECTASHIELD Antifade Mounting Medium with DAPI (Ref. H-1200; Vector Laboratories). We used mouse anti-$\alpha$-tubulin (1:200, Ref. T8203; Sigma-Aldrich) as primary antibodies and Cy3-conjugated donkey anti-mouse IgG (1:150, Ref. 715-165-151; Jackson ImmunoResearch) as secondary antibodies. Incubation with Alexa Fluor 488–conjugated phalloidin (10 U/ml, Ref. A12379; Thermo Fisher Scientific) was performed along with secondary antibody incubation.

### EU labeling of nascent oocyte RNAs
As in Reference 23, oocytes arrested in prophase I and isolated from follicular cells were cleared of the *zona pellucida* in M2 + BSA, and milrinone medium with 0.4% pronase. EU incorporation and detection were performed using Click-iT RNA Alexa Fluor 488 Imaging Kit (Ref. C10329; Thermo Fisher Scientific). Oocytes were incubated in M2 + BSA, and milrinone medium with or without 0.5 mM EU for 3 h at 37°C and fixed in 1X PBS and 4% paraformaldehyde for 30 min at 37°C. Oocytes were subsequently washed in 1X PBS where they were stored overnight at 4°C. Oocyte permeabilization and washes were

done at room temperature in 1X PBS and 0.5% Triton X-100 for 10 min and in 1X PBS, respectively. EU detection was performed as indicated by the manufacturer, and oocytes were then washed twice in Click-iT reaction rinse buffer and mounted on slide wells filled with VECTASHIELD Antifade Mounting Medium with DAPI (Ref. H-1200; Vector Laboratories).

## Imaging

### Spinning disk microscopy
Images were acquired with a PlanAPO 40×/1.25 NA objective on a Leica DMI6000B microscope enclosed in a thermostatic chamber (Life Imaging Service) equipped with a Retiga 3 CCD camera (QImaging) coupled to a Sutter filter wheel (Roper Scientific) and a Yokogawa CSU-X1-M1 spinning disk. Data were collected with MetaMorph software (Universal Imaging).

### STED microscopy
STED imaging was performed with a STEDYCON module from Abberior mounted on a Zeiss Axio Observer 7 inverted video microscope coupled to a sCMOS camera (Hamamatsu Flash 4) and a PlanAPO 100× oil-immersion objective (NA = 1.46) for an effective pixel size of 50 nm (Zeiss).

### OMX microscopy
Structured illumination (3D-SIM) was performed using a DeltaVision OMX Blaze microscope (GE Healthcare) coupled to a Photometrics Evolve 512 EMCCD camera (Photometrics) and a PlanAPO 60× oil-immersion objective (NA = 1.4) for an effective pixel size of 80 and 40 nm after reconstruction (Olympus).

### Light-sheet fluorescence microscopy
The cleared samples are acquired with the ultramicroscope I (LaVision BioTec-Miltenyi) assisted by ImSpector Pro software (LaVision BioTec-Miltenyi). The light sheet is created by two cylindrical lenses using the laser with three specific wavelengths: 561, 640, and 785 nm (Coherent Sapphire Laser, LaVision BioTec-Miltenyi). For all the acquisitions, we used a binocular stereomicroscope (MVX10, Olympus) with a 2× objective (MVPLAPO, Olympus) adding a protective dipping cap with corrective lens. For detection, we used a Zyla sCMOS camera (2,048 × 2,048 pixels) from Andor Oxford Instruments. The images were in 16-bit. We used for zoom magnification 3.2X (pixel sizes x,y = 1.02 $\mu$m). Samples were dived in a quartz tank (LaVision BioTec-Miltenyi) filled with DBE to maintain the clearing. The z-step size between each image was 1 $\mu$m with 150-ms time exposure.

## Oocyte RNA extraction

Total RNA extraction was performed as in Reference 16 from fully grown oocytes with the *zona pellucida*. For the RNA-Seq analysis, three biological replicates were performed each containing 10 oocytes from two $Myo10^{flox/flox}$; $Cre^+$ mice ($Myo10^{-/-}$ full oocytes) and two $Myo10^{wt/flox}$; $Cre^+$ mice ($Myo10^{+/-}$ oocytes). For the RT–qPCR analysis, two biological replicates were made containing 30 and 24 oocytes from two $Myo10^{flox/flox}$; $Cre^-$ mice ($Myo10^{-/-}$ full oocytes) and two $Myo10^{wt/flox}$; $Cre^+$ mice ($Myo10^{+/-}$ oocytes). Total RNA extraction

was performed using RNAqueous-Micro Total RNA Isolation Kit (Ref. AM1931; Thermo Fisher Scientific). Briefly, oocytes were washed three times in 1X PBS, transferred to lysis buffer (supplemented with 1 $\mu$l $\beta$-mercaptoethanol for the RNA-Seq analysis), and frozen in liquid nitrogen where they were stored at –80°C for 1 h. After thawing, RNA extraction was performed as described by the manufacturer. RNAs were eluted twice in 10 $\mu$l elution buffer and treated with DNaseI.

## RNA-seq and data analysis

### cDNA libraries and RNA-seq
Library preparation and Illumina sequencing were performed at the Ecole normale supérieure Genomics Core Facility. 5 ng of total RNA was amplified and reverse-transcribed to cDNA using SMART-Seq v4 Ultra Low Input RNA Kit (Ref. 634889; Clontech). Afterward, an average of 500 pg of amplified cDNA was used to prepare the libraries following the Nextera XT DNA Kit (Ref. FC-131-1096; Illumina). Libraries were multiplexed by 6 on high-output flow cells. A 75-bp read sequencing was performed on a NextSeq 500 device (Illumina). A mean of 26 ± 4 million passing Illumina quality filter reads was obtained for each of the six samples. The sequences can be found on the European Nucleotide Archive website, accession number PRJEB56821.

### Bioinformatics analysis
The analysis was performed using the Galaxy server of the ARTbio Bioinformatics Platform. Sequencing quality control was carried out with FastQC (Babraham Bioinformatics) (Galaxy version 0.72+-galaxy1) and MultiQC tools (v1.6). Alignment of reads to the genome of *Mus musculus* mm10 reference genome was done with Bowtie 2 (v2.3.4.2) (54) with fast local option for soft clipping of the Nextera transposase adapter sequence. We used the mm10 GTF file (Ensembl GTF Mus_musculus.GRCm38.94.chr) as annotation files. Counting of reads per gene and differential gene expression analysis were performed with featureCounts (v1.6.3+galaxy2) and DESeq2 (v2.11.40.4), respectively. Transcripts were determined as differentially expressed based on a *P*-adj (adjusted *P*-value with false discovery rate correction) of less than 0.05. Of note, 222 deregulated transcripts were annotated as predicted (32.4% of the deregulated transcripts; Table S1) and were predominantly pseudogenes (84.2% of deregulated predicted transcripts). The Galaxy workflows can be found on our GitHub repository (https://github.com/Terretme/Myosin10-full-KO-RNAseq).

### Gene ontology (GO) enrichment analysis
The analysis was performed with the PANTHER Classification System Overrepresentation Test (Released 20210224), with the annotation dataset GO biological process complete, using Fisher's exact test with a *P*-value with a false discovery rate correction threshold of 0.05. For the GO biological process complete (Table S1), the deregulated transcript list from the RNA-Seq analysis was used as the input list and the *Mus musculus* genome as the reference list. For the graph in Table S1, to provide an overview of the different enriched pathways, the enriched GO terms were selected as follows: for each term with a fold enrichment >10, we browsed their ancestor chart using QuickGO, and selected the highest ancestor

term (only related by the type "A is a B") that is among the enriched terms with a fold enrichment >10, if any. If no ancestor was found with these criteria, the initial enriched term was mounted in a graph. The selected GO terms are marked in blue in Table S1.

## RT–qPCR

Reverse transcription of the extracted RNAs was performed using iScript Reverse Transcription Supermix (Ref. 1708840; Bio-Rad). We used SsoAdvanced Universal SYBR Green Supermix (Ref. 1725270; Bio-Rad) and CFX-96 (Bio-Rad) for quantitative PCR of cDNAs using the primers shown below. Two technical replicates were made for each condition. mRNA quantity was normalized to *Gapdh*, and relative levels of mRNA expression were calculated with the 2-ΔΔC. The mean and the SEM were normalized to $Myo10^{+/-}$. The selected genes (see list below) are marked in green in Table S1.

**Selected genes to validate the RNA-Seq by RT-qPCR.**

|          | Forward primer           | Reverse primer           |
|----------|--------------------------|--------------------------|
| *Gapdh*  | TGGAGAAACCTGCCAAGTATG     | GGTCCTCAGTGTAGCCCAAG     |
| *Akr1b3* | AGAGCATGGTGAAAGGAGCC      | CCATAGCCGTCCAAGTGTCC     |
| *Ccnb1*  | GAGAGGTTGACGTCGAGCAG      | GAGTTGGTGTCCATTCACCG     |
| *Eif4e*  | GAGGTTGCTAACCCAGAGCA      | CATAGGCTCAATCCCGTCCTT    |
| *Hsp90aa1* | CGTCTCGTGCGTGTTCATTC    | CCAGAGCGTCCGATGAATTG     |
| *Myo10*  | GGCACGAAAGCAATATAGAAAGG   | CTTCTGGAACACGATGGCTG     |
| *Oaz1*   | CAGCAGCGAGAGTTCTAGGG      | CTCGTCGGAGTACAGGATGC     |
| *Oser1*  | CCAGCTGCGAGGAGTATTAAC     | GGGCCTTCTCCAACAGACAG     |
| *Paox*   | GGTCCAGCCTTCTTTTGAGTCT    | GTAGGAACCTCGGGTGTACG     |
| *Psma5*  | CAGGTGCTATGTCTCGTCCC      | ATGAGCGAGGACTTGATGGC     |
| *Rpl19*  | CGCTGCGGGAAAAAGAAGGT      | CCTCTTCCCTATGCCCATATGC   |
| *Rps4x*  | GGCGAGTCTCTTTCCGTTCC      | GGATGGACGAGGAGCAAACAC    |
| *Stxbp5l* | GATTTGCAAGACAGTTCGGC     | TGGCAGTAACAGTCAACAC      |

## Machine-learning approach for oocyte morphological characterization

We used our machine-learning pipeline presented in Reference 21 to automatically measure and compare the morphology of $Myo10^{-/-}$ and control oocytes from brightfield images. Oocyte and *zona pellucida* contours were automatically determined with Oocytor neural networks. This segmentation allowed us to extract the value of 89 features for each oocyte–*zona pellucida* complex. These features characterize the size, shape, global intensity repartition, local texture, and spatial organization of the oocyte, perivitelline space, and *zona pellucida*. The complete list and description of the defined features are given in our GitHub repository (https://github.com/gletort/Oocytor). As the features are not independent between themselves, we automatically selected a subset of uncorrelated features (with a threshold of 0.75) as described in Reference 21. We next trained a Random Forest classifier (55) with

these features to distinguish between fully grown $Myo10^{-/-}$ and control oocytes cleared of follicular cells. Importantly for our study, the Random Forest algorithm scored each feature according to its importance for classification, based on the Gini index. We checked that the algorithm was able to recognize the oocyte type with a 10-fold cross-validation on our dataset and obtained a classification accuracy of 82% (with a precision rate of 80.9% and a recall rate of 83%). The most important features for classification concerned the composition of the *zona pellucida* with the presence of vertical linear structures in it (TZP-like structure), the texture of the cytoplasm, the heterogeneity in the thickness of the perivitelline space, and the circularity of both the oocyte and the internal limit of the *zona pellucida*. Manual inspection of these features suggested that the difference in the texture of the cytoplasm was mainly due to differences in nucleolus focus rather than actual morphological differences, and this feature was not statistically different between $Myo10^{-/-}$ full and control oocytes during meiotic maturation. We therefore decided not to explore this characteristic further.

### *Description of the features shown in this study*
**Vertical linear structure in the *zona pellucida*** To quantify the presence of vertical linear structures resembling TZPs, the area in the segmented *zona pellucida* was first flattened with the " Flatten " option in Fiji (NIH) to obtain a horizontal rectangle. Linear structures were then detected by measuring and thresholding the " tubeness " of each pixel in the image (with " Tubeness " plugin in Fiji, which is based on the eigenvalues of the Hessian matrix of the image). Finally, only fairly vertical structures were kept by applying Sobel filters on the thresholded images.

**Perivitelline space thickness** Thickness was calculated as the distance between the oocyte shape and the internal limit of the *zona pellucida* at each angle around the oocyte center. The mean, minimum, maximum, SD (data not shown), and coefficient of variation were then calculated from the distribution of distances.

**Oocyte aspect ratio** Aspect ratio was calculated as the ratio between the maximum Feret diameter (longest distance) of the oocyte shape and the minimum Feret diameter (distance in the perpendicular direction to the maximum Feret diameter). A circle has an aspect ratio of 1, whereas an elongated shape has a higher aspect ratio.

## Image analysis

For all signal intensity measurements, the image background was subtracted using MetaMorph software.

### *Signal intensity of SiR-actin–labeled TZPs*
Measurements were performed on growing and fully grown oocytes cleared of follicular cells and labeled with SiR-actin. For each oocyte, the fifth image among eight Z acquisitions spaced 4 µm apart was selected for measurement, which corresponded to the oocyte equatorial plane. After background subtraction, we drew an ellipsoidal ROI of three pixels (0.34 µm) width along the pericenter of the *zona pellucida* using Fiji. The mean gray value of SiR-actin signal in the ROI was measured as a readout of TZP density. For

each oocyte, the mean gray value was normalized to that of the control oocyte with the highest mean gray value. The diameter of each oocyte was measured in parallel.

### Signal intensity of EU-labeled RNAs

Measurements were performed using MetaMorph similar to Reference 23 on growing and fully grown oocytes. After background subtraction and sum of Z projection of the DAPI and EU signal images (images were spaced 1 $\mu m$ apart and selected for Z projection based on the DAPI signal), the integrated intensity of the DAPI and EU signals was measured in a ROI of 77,577 pixels (999 $\mu m^2$). The same ROI was used for all measurements and was drawn to encompass the DAPI signal of each oocyte. The integrated intensity of the EU signal was then normalized to that of the DAPI signal. For each oocyte, the relative EU intensity value was normalized to that of the control oocyte with the highest relative EU intensity value.

### Signal intensity of phalloidin-labeled cytoplasmic F-actin

Measurements were done on oocytes fixed 6 h after NEBD using MetaMorph as in Reference 16. After background subtraction, the integrated intensity of the phalloidin signal was measured in a ROI containing the cytoplasm of the oocyte equatorial plane. For each oocyte, we used the same ROI of 216,290 pixels (2,786 $\mu m^2$) and the integrated intensity of the phalloidin signal was normalized to that of the control oocyte with the highest value of phalloidin signal intensity.

### Ovarian follicular density

Cleared ovaries were imaged by light-sheet fluorescence microscopy, and ovarian slices, one every 50 slices, were selected for measurement (slices were spaced 1 $\mu m$ apart). For each image, the area of the ovarian slice was measured by thresholding the TO-PRO-3 signal using Fiji. Briefly, images were preprocessed with a gamma and a median filter and thresholded with the Otsu dark method. When image thresholding led to more than one ovarian area because of discontinuous signal, only the largest thresholded area was selected. For each image, we counted the number of follicles in the thresholded ovarian slice. Follicles were determined based on a defined outline, with an apparent oocyte surrounded by visible follicular cell layers (this includes preantral and antral follicles). The number of follicles in the thresholded ovarian slice was then normalized to the slice area. The follicular density of the ovary corresponds to the mean of the follicular density of the slices analyzed.

### Timing of polar body extrusion

The timing of the first polar body extrusion was determined in oocytes monitored by live spinning disk microscopy. Only videos with control oocytes extruding a polar body before 12 h after NEBD were selected.

### Spindle bipolarization and positioning

Spindle morphogenesis and positioning were assessed in SiR-tub–labeled oocytes monitored during meiotic maturation by live spinning disk microscopy. The timing of spindle bipolarization was assessed by scoring the percentage of oocytes with a bipolarized spindle every hour from 1 to 6 h after NEBD. Spindle interpolar

length and central width were measured 7 h after NEBD using MetaMorph. Only spindles parallel to the acquisition plane were measured. Spindle localization was assessed by scoring the percentage of oocytes with a central or off-centered spindle 30 min before anaphase for oocytes that extruded a polar body or up to 12 h after NEBD for those that did not.

### Chromosome alignment

Metaphase plate height and width were measured as in Reference 16 using MetaMorph in H2B-GFP–expressing oocytes monitored during meiotic maturation by live spinning disk microscopy. We measured the height and width of a ROI rectangle enclosing chromosomes 30 min before anaphase for oocytes that extruded a polar body or 12 h after NEBD for those that did not. Only metaphase plates oriented parallel to the acquisition plane were measured.

### Cumulus–oocyte coupling

Measurements were performed on cumulus–oocyte complexes when single follicular cells were visible, still connected to the oocyte by TZPs (visible with SiR-actin). The coupling was quantified for each follicular cell by measuring the ratio of Lucifer Yellow fluorescence intensity between the follicular cell and the oocyte 30 min after Lucifer Yellow microinjection into the oocyte. For that, after acquiring the whole cumulus–oocyte complex (25 Z acquisitions spaced 5 $\mu m$ apart), we selected the plane where a single follicular cell connected to the oocyte via TZPs was visible. We drew an ellipsoidal ROI of 19 pixels width and 16 pixels height (4.3 $\mu m$ × 3.6 $\mu m$) centered on the follicular cell using Fiji. The mean gray value of Lucifer Yellow signal in the ROI was measured. We displaced the ROI to put it in the oocyte and measured the mean gray value of Lucifer Yellow signal in the ROI. The mean gray value of Lucifer Yellow signal in the follicular cell was then divided by the one in the oocyte. This was repeated for all single follicular cells.

## Statistical analysis

We used GraphPad Prism 7.0a software for statistical analysis of RT–qPCR, oocyte polar body extrusion rate, and image data. The Gaussian distribution of values was tested using a D'Agostino & Pearson normality test. Comparison of means was performed by a two-tailed unpaired $t$ test (with Welch's correction for unequal variances) or by a two-tailed Mann–Whitney test (for a non-Gaussian distribution of values). For comparison of more than two conditions, a ANOVA test comparing the mean of each column with the mean of every other column when the data followed a Gaussian distribution, and a Kruskal–Wallis test comparing the mean rank of each column with the mean rank of every other column when the data did not were performed. We used two-sided Fisher's exact tests and a chi-squared test for contingency table analysis. All tests were performed with a confidence interval of 95%, n.s (not significant), $P \geq 0.05$, *$P < 0.05$, **$P < 0.01$, ***$P < 0.001$, and ****$P < 0.0001$. For each graph, the error bars indicate the SD or the SEM.

## Ethical statement

All experimental procedures used for the project have been approved by the Ministry of Agriculture to be conducted in our CIRB

Animal Facility (authorization no. 75-1170). The use of all the genetically modified organisms described in this project has been granted by the DGRI (Direction Générale de la Recherche et de l'Innovation: Agrément OGM; DUO-5291).

## Data Availability

For the RNA-Seq, the sequences can be found on the European Nucleotide Archive website, accession number PRJEB56821. The Galaxy workflows of the bioinformatics analysis of the RNA-Seq can be found on our GitHub repository (https://github.com/Terretme/Myosin10-full-KO-RNAseq).

## Supplementary Information

## Acknowledgements

We thank the Verlhac/Terret laboratory for comments on the article and discussion; and C Antoniewski and N Naouar from the IBPS ARTbio Bioinformatics Platform for their services and discussion. The STED microscopy was performed at the Orion Platform (member of France-Bioimaging ANR-10-INBS-XX) of the Center for Interdisciplinary Research in Biology (UMR7241/U1050) of Collège de France. The OMX microscopy was performed at the Cell and Tissue Imaging Platform-PICT-IBiSA (member of France-Bioimaging ANR-10-INBS-04) of the Genetics and Developmental Biology Department (UMR3215/U934) of Institut Curie, supported by the European Research Council (ERC EPIGENETIX N°250367). The work at the IBENS Genomics Platform was supported by the France Génomique national infrastructure, funded as part of the "Investissements d'Avenir" program managed by the Agence Nationale de la Recherche (reference: ANR-10-INBS-09). This work was supported by the Fondation pour la Recherche Médicale (FRM Label EQU201903007796 to M-H Verlhac), by the Agence Nationale de la Recherche (ANR-18-CE13 to M-H Verlhac and Auguste Genovesio, IBENS/ENS, ANR-16-CE13 to M-E Terret), and by the France Canada Research Fund (FCRF 2017 to M-E Terret and Hugh J Clarke, McGill University). F Crozet obtained a grant from the Fondation pour la Recherche Médicale for her fourth PhD year (FDT202001010906). This work has received support from the Fondation Bettencourt Schueller, and support under the program "Investissements d'Avenir" launched by the French Government and implemented by the Agence Nationale de la Recherche, with the references: ANR-10-LABX-54 MEMO LIFE and ANR-11-IDEX-0001-02 PSL* Research University.

### Author Contributions

F Crozet: conceptualization, formal analysis, investigation, methodology, and writing—original draft, review, and editing.
G Letort: data curation, software, formal analysis, investigation, methodology, and writing—review and editing.
R Bulteau: formal analysis, investigation, methodology, and writing—review and editing.
C Da Silva: formal analysis, investigation, methodology, and writing—review and editing.
A Eichmuller: formal analysis and investigation.
A-F Tortorelli: formal analysis, investigation, and writing—review and editing.
J Blévinal: formal analysis, investigation, and methodology.
M Belle: formal analysis, investigation, methodology, and writing—review and editing.
J Dumont: investigation, methodology, and writing—review and editing.
T Piolot: investigation, methodology, and writing—review and editing.
A Dauphin: investigation, methodology, and writing—review and editing.
F Coulpier: investigation, methodology, and writing—review and editing.
A Chédotal: methodology.
J-L Maître: methodology and writing—review and editing.
M-H Verlhac: conceptualization, funding acquisition, and writing—review and editing.
HJ Clarke: funding acquisition and writing—review and editing.
M-E Terret: conceptualization, data curation, formal analysis, supervision, funding acquisition, investigation, methodology, and writing—review and editing.

### Conflict of Interest Statement

The authors declare that they have no conflict of interest.

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
