## [Reviewer comments · Life Science Alliance]

Life Science Alliance

filopodia-like protrusions of adjacent somatic cells shape the developmental potential of oocytes

Flora Crozet, Gaelle Letort, Rose Bulteau, Christelle Da Silva, adrien eichmuller, Anna-Francesca Tortorelli, Joséphine Blévinat, Morgane Belle, Julien Dumont, Tristan Pilot, Aurelien Dauphin, Fanny Couplier, Alain Chedotal, Jean-Léon Maitre, Marie-Helene Verlhac, Hugh Clarke, and Marie-Emilie Terret

DOI: <https://doi.org/10.26508/lsa.202301963>

Corresponding author(s): Marie-Emilie Terret, Center for Interdisciplinary Research in Biology

Review Timeline:

Submission Date:	2023-02-01
Editorial Decision:	2023-02-01
Revision Received:	2023-02-06
Editorial Decision:	2023-02-27
Revision Received:	2023-03-07
Accepted:	2023-03-07

Transaction Report:

Please note that the manuscript was reviewed at Review Commons and these reports were taken into account in the decision-making process at Life Science Alliance.

Manuscript number: RC-2022-01672

Corresponding author(s): Marie-Emilie TERRET

1. General Statements [optional]

We thank the three reviewers for carefully evaluating our study and for their constructive comments and suggestions. We are also grateful for their clearly supportive appreciation of our work, and for noting that it could have a broad impact in several fields of biology.

2. Description of the planned revisions

Requests from all reviewers:

#1 All reviewers asked whether cumulus-oocyte coupling was affected in *Myo10* full knockouts, as they have significantly fewer transzonal projections (TZPs) between the oocyte and the surrounding somatic cells (Fig.1 E). They all proposed dye diffusion assays, *i.e.*, microinjection of a fluorophore into the oocyte to visualize if it spreads into the surrounding somatic cells. We conducted this experiment (see Fig.A below), injecting Lucifer Yellow into Control and *Myo10*^{-/-} full oocytes. In addition, we treated Control oocytes with carbenoxolone (CBX), a gap-junction blocker, to prevent the transfer of Lucifer Yellow from the oocyte to the surrounding somatic cells via gap-junctions, as a negative control. Our results show that some connectivity remains in *Myo10* full knockouts, which could explain the fact that these oocytes do not escape the prophase arrest. We are quantifying now the amount of connectivity.

Fig.A: Control and *Myo10*^{-/-} full live cumulus-oocyte complexes stained with SiR-actin (magenta) to label TZPs in which each oocyte was injected with 10% Lucifer Yellow (green). CBX-treated oocytes were incubated for 1h in 100 μ M carbenoxolone to block gap-junctions. Scale bar 20 μ m.

#2 All reviewers asked why there is an apparent discrepancy between the results for detecting the numbers and densities of TZPs. This is because we did not explain the phenotype well enough and we apologize for this. There is in fact no contradiction between the number and density of TZPs in *Myo10* full knockouts in the manuscript. The TZP-like structure that are visible in

Revision Plan

transmitted light in the *Myo10* full knockouts always contain actin (see Fig.B below). We see less TZPs in *Myo10* full knockouts with fluorescence-based methods (Fig.1 E), but most of these remaining TZPs are visible by transmitted light. In Control oocytes, TZPs are abundant, as evidenced with fluorescence-based methods (Fig.1 E), but not visible by transmitted light (Fig.2 F and Fig.B above). These data were missing from our initial submission and will be added in our revised manuscript.

Fig.B: Cropped images of *zona pellucida* from Control and *Myo10*^{-/-} full oocytes. Brightfield images on the left panels, SiR-actin (white) labeled TZPs on the middle panels, merge (SiR-actin in magenta) on the right panels. Scale bar 9 μ m.

Our observations suggest that in *Myo10* full knockouts, the structure of the remaining TZPs and/or that of the *zona pellucida* is different from that of Control oocytes. Our hypothesis, supported by the observation that we occasionally observe some follicular-like cells located ectopically within the perivitelline space of *Myo10* full knockouts (Fig.S4 C) and that the texture of the *zona pellucida* is different between Control and *Myo10* full knockouts (Fig.2 C), is that the properties of the *zona pellucida* (a viscoelastic extracellular matrix) might change as a function of the crosslinking generated by the number of TZPs passing through it. If so, this could change its mechanical properties. Reviewer 2 proposed to perform electron microscopy to study these TZP-like structures that he thought lacked actin. Since this is not the case (see Fig.B above), and since our hypothesis is that the *zona pellucida* might have different properties, we conducted

experiment to assess the mechanical properties of the *zona pellucida* of Control and *Myo10* full knockouts using Atomic Force Microscopy (AFM). Our experiments show a decreased *zona pellucida* elasticity in *Myo10* full knockouts (see Fig.C below). It suggests that the decrease in crosslinking generated by the reduced number of TZPs passing through it affects the mechanical properties of the *zona pellucida*.

Fig.C: Elasticity of the *zona pellucida* of Control and *Myo10*^{-/-} full oocytes, measured with AFM. Data are from three independent experiments (28 Control and 27 *Myo10*^{-/-} full oocytes). Statistical significance of differences was assessed by a two-tailed Mann Whitney test, P value = 0.0098.

Specific requests from Reviewer 1:

#1 The reviewer suggested measuring cGMP and cAMP into oocytes during meiotic arrest and resumption to test if there is a direct blockage in the transfer of molecules from follicular cells to the oocyte. We will not perform this experiment, as it is not feasible due to the large number of oocytes that would be required. However, the dye diffusion assays requested by the three reviewers will give the same type of information.

#2 We will follow all of the reviewer's suggestions by 1/ rewording (line 53-55), 2/ explaining better (lines 145-153: two objects of the same size Fig.2 B can have a different perimeter Fig.S4 F if they do not have the same shape, which is the case here, see aspect ratio Fig.S4 E), and 3/ citing the following paper (Zhang et al., 2021; Nature Comm., 12:2523) that we did not cite in the first version of our manuscript because their oocyte-specific *Radixin* knockout causes a loss of oocyte microvilli, and a decrease of TZPs, causing subfertility. Because oocyte-specific *Radixin* knockout alters both oocyte microvilli and TZPs, it is impossible to clearly distinguish the contribution of decreased TZPs to subfertility.

Specific requests from Reviewer 2:

#1 The reviewer asked to repeat our transparization experiment with more animals, and we did so. We added 3 mice for each group for a total of 5 mice per group, which is above standards. We are currently analyzing the results. However, we believe that the number of follicles per unit area is the relevant parameter for fertility, since it overcomes the physiological variability in ovarian size. We have plotted below the weight of the ovaries of each animal (24 Control mice and 27 *Myo10* full knockouts), light and dark grey for the ovaries of the same Control mouse, light

Revision Plan

and dark green for the ovaries of the same *Myo10* full knockout. It is very clear that in most cases, the weight of the ovaries is very variable in the same mouse, and can go from simple to double.

#2 We will follow all of the reviewer's suggestions by 1/ discussing perivitelline space formation (not delayed but rather precocious, see Fig.2 E), 2/ discussing GO pathways more in depth, and 3/ making comparisons between mouse strains using ANOVA for Fig.1 E, Fig.2 B, Fig.2 C, Fig.2 E, Fig.2 G, Fig.S4 A, Fig.S4 B, Fig.S4 E, Fig.S4 F. In fact, we have already done these (see below, old graphs paired with new statistics) and find, as expected, that all Controls and the oocyte-specific knockout are not significantly different from each other, whereas the whole-body knockout is significantly different from all of them. We will include these new statistics in our revised manuscript. 4/ At last, we will tone down the 5-ethynyl uridine experiment.

ANOVA tests comparing the mean of each column with the mean of every other column when the data followed a Gaussian distribution, and Kruskal-Wallis tests comparing the mean rank of each column with the mean rank of every other column when the data did not.

Fig.1 E

Ctrl full vs. Myo10 ^{-/-} full	****	<0,0001
Ctrl full vs. Ctrl oocyte	ns	0,6857
Ctrl full vs. Myo10 ^{-/-} oocyte	ns	0,4479
Myo10 ^{-/-} full vs. Ctrl oocyte	****	<0,0001
Myo10 ^{-/-} full vs. Myo10 ^{-/-} oocyte	****	<0,0001
Ctrl oocyte vs. Myo10 ^{-/-} oocyte	ns	0,9722

Revision Plan

Fig.2 B

Ctrl full vs. $Myo10^{-/-}$ full	ns	0,4459
Ctrl full vs. Ctrl oocyte	ns	0,7355
Ctrl full vs. $Myo10^{-/-}$ oocyte	ns	>0,9999
$Myo10^{-/-}$ full vs. Ctrl oocyte	ns	0,9905
$Myo10^{-/-}$ full vs. $Myo10^{-/-}$ oocyte	ns	>0,9999
Ctrl oocyte vs. $Myo10^{-/-}$ oocyte	ns	>0,9999

Fig.2 C

Ctrl full vs. $Myo10^{-/-}$ full	*	0,0280
Ctrl full vs. Ctrl oocyte	ns	0,3355
Ctrl full vs. $Myo10^{-/-}$ oocyte	ns	>0,9999
$Myo10^{-/-}$ full vs. Ctrl oocyte	****	<0,0001
$Myo10^{-/-}$ full vs. $Myo10^{-/-}$ oocyte	**	0,0083
Ctrl oocyte vs. $Myo10^{-/-}$ oocyte	ns	0,3895

Fig.2 E

Ctrl full vs. $Myo10^{-/-}$ full	****	<0,0001
Ctrl full vs. Ctrl oocyte	ns	>0,9999
Ctrl full vs. $Myo10^{-/-}$ oocyte	ns	0,3139
$Myo10^{-/-}$ full vs. Ctrl oocyte	****	<0,0001
$Myo10^{-/-}$ full vs. $Myo10^{-/-}$ oocyte	*	0,0103
Ctrl oocyte vs. $Myo10^{-/-}$ oocyte	ns	0,0811

Fig.2 G

Ctrl full vs. $Myo10^{-/-}$ full	****	<0,0001
Ctrl full vs. Ctrl oocyte	ns	0,7183
Ctrl full vs. $Myo10^{-/-}$ oocyte	ns	0,7291
$Myo10^{-/-}$ full vs. Ctrl oocyte	****	<0,0001
$Myo10^{-/-}$ full vs. $Myo10^{-/-}$ oocyte	****	<0,0001
Ctrl oocyte vs. $Myo10^{-/-}$ oocyte	ns	0,9999

Revision Plan

Fig.S4 A

Ctrl full vs. Myo10 ^{-/-} full	ns	>0,9999
Ctrl full vs. Ctrl oocyte	ns	0,0503
Ctrl full vs. Myo10 ^{-/-} oocyte	ns	0,0566
Myo10 ^{-/-} full vs. Ctrl oocyte	**	0,0063
Myo10 ^{-/-} full vs. Myo10 ^{-/-} oocyte	*	0,0388
Ctrl oocyte vs. Myo10 ^{-/-} oocyte	ns	>0,9999

Fig.S4 B

Ctrl full vs. Myo10 ^{-/-} full	ns	>0,9999
Ctrl full vs. Ctrl oocyte	ns	0,9997
Ctrl full vs. Myo10 ^{-/-} oocyte	ns	0,5007
Myo10 ^{-/-} full vs. Ctrl oocyte	ns	0,9941
Myo10 ^{-/-} full vs. Myo10 ^{-/-} oocyte	ns	0,9522
Ctrl oocyte vs. Myo10 ^{-/-} oocyte	ns	0,8581

Fig.S4 E

Ctrl full vs. Myo10 ^{-/-} full	**	0,0030
Ctrl full vs. Ctrl oocyte	ns	0,4773
Ctrl full vs. Myo10 ^{-/-} oocyte	ns	0,4353
Myo10 ^{-/-} full vs. Ctrl oocyte	ns	>0,9999
Myo10 ^{-/-} full vs. Myo10 ^{-/-} oocyte	ns	>0,9999
Ctrl oocyte vs. Myo10 ^{-/-} oocyte	ns	>0,9999

Fig.S4 F

Ctrl full vs. Myo10 ^{-/-} full	*	0,0412
Ctrl full vs. Ctrl oocyte	ns	0,6078
Ctrl full vs. Myo10 ^{-/-} oocyte	ns	>0,9999
Myo10 ^{-/-} full vs. Ctrl oocyte	ns	0,9516
Myo10 ^{-/-} full vs. Myo10 ^{-/-} oocyte	ns	>0,9999
Ctrl oocyte vs. Myo10 ^{-/-} oocyte	ns	>0,9999

Specific requests from Reviewer 3:

#1 In Fig.2 B, the data set does not differ (ns, see old statistics in the Figure Legends and new statistics above).

#2 We will not *in vitro* fertilize superovulated *Myo10* full knockouts to check if sperm is bound to the *zona pellucida* or within the perivitelline space, since our *in vivo* matings (Fig.6 A) show that *Myo10* full knockouts can be fertilized, albeit less efficiently than Controls, suggesting that the *zona pellucida* is at least partially functional. While we recognize that this is an interesting question, we believe that it could be the object of a future study, more focused on the *zona pellucida*.

3. Description of the revisions that have already been incorporated in the transferred manuscript

The current manuscript is not different from the original submission.

4. Description of analyses that authors prefer not to carry out

Reviewer 1 suggested measuring cGMP and cAMP into oocytes during meiotic arrest and resumption to test if there is a direct blockage in the transfer of molecules from follicular cells to the oocyte. We will not perform this experiment, as it is not feasible due to the large number of oocytes that would be required. However, the dye diffusion assays requested by the three reviewers will give the same type of information.

We will not *in vitro* fertilize superovulated *Myo10* full knockouts to check if sperm is bound to the *zona pellucida* or within the perivitelline space, since our *in vivo* matings (Fig.6 A) show that *Myo10* full knockouts can be fertilized, albeit less efficiently than Controls, suggesting that the *zona pellucida* is at least partially functional. While we recognize that this is an interesting question, we believe that it could be the object of a future study, more focused on the *zona pellucida*.

It should be noted that we almost lost this line during the various Covid-related confinements, where up to 60% of the animals in our animal facilities had to be sacrificed. We have very few mice of the right genotype to work with (1/8th of the animals generated by flox/wt couples are flox/flox females, with an average of 7 live animals per litter for one litter per couple per month), and few oocytes per mouse (around 10 oocytes per female in *Myo10* knockouts, Fig. 6A).

February 1, 2023

Re: Life Science Alliance manuscript #LSA-2023-01963-T

Dr. Marie-Emilie Terret
Collège de France
CIRB
11 place Marcelin Berthelot
Paris 75005
France

Dear Dr. Terret,

Thank you for submitting your manuscript entitled "The filipodia-like protrusions of adjacent somatic cells shape the developmental potential of mouse oocytes" to Life Science Alliance. We invite you to re-submit the manuscript, revised according to your Revision Plan.

Thank you for this interesting contribution to Life Science Alliance. We are looking forward to receiving your revised manuscript.

Sincerely,

Eric Sawey, PhD
Executive Editor
Life Science Alliance
<http://www.lsa-journal.org>

B. MANUSCRIPT ORGANIZATION AND FORMATTING:

We thank the three reviewers for carefully evaluating our study and for their constructive comments and suggestions. We are also grateful for their clearly supportive appreciation of our work, and for noting that it could have a broad impact in several fields of biology. Please note that all the text modifications are in blue in the revised manuscript.

Reviewer 1:

Evidence, reproducibility and clarity (Required):

In the manuscript by Crozet et al. the authors investigated the contribution of transzonal projections (TZPs) to the oocyte development and acquisition of competence. The results were obtained using two Myo10 knockout mice models: a full knockout for Myo10 (Myo10^{-/-} full) and an oocyte-conditioned knockout (Myo10^{-/-} oo). The major findings due to the global depletion of Myo10 include the decrease in TZP density, discrete morphological alterations in the oocytes, alterations in oocyte gene expression, the inability of the oocytes to complete the first meiotic division (lack of 1PB extrusion), and subfertility in Myo10^{-/-} full females.

The research topic is interesting and, overall, I consider the manuscript relevant.

We thank the reviewer for his/her encouraging comments.

However, to increase the scientific soundness authors are encouraged to explore the effects of the (partial) interruption of the germ-soma communication on the regulation of meiotic arrest and resumption. This is worth investigating (is optional, but highly recommended) since the lower density of TZPs is associated with an apparent normal meiotic arrest but an abnormal meiotic resumption. At first, the measurement of cGMP and cAMP into oocytes during meiotic arrest and resumption would be a nice try. This will help to shed light on the reasons for the abnormal meiotic progression, indicating if it is the consequence of a direct blockage in the transfer of molecules from follicular cells to the oocyte or an indirect consequence.

The reviewer suggested measuring cGMP and cAMP into oocytes during meiotic arrest and resumption to test if there is a direct blockage in the transfer of molecules from follicular cells to the oocyte. We will not perform this experiment, as it is not feasible due to the large number of oocytes that would be required. However, the dye diffusion assays requested by the three reviewers gives the same type of information (see below).

It should be noted that we almost lost this line during the various Covid-related confinements, where up to 60% of the animals in our animal facilities had to be sacrificed. We have very few mice of the right genotype to work with (1/8th of the animals generated by flox/wt couples are flox/flox females, with an average of 7 live animals per litter for one litter per couple per month), and few oocytes per mouse (around 10 oocytes per female in Myo10 knockouts, Fig. 6A).

Minor points

- Lines 53-55: The oocyte does not complete two successive meiotic divisions to generate a mature oocyte ready to be fertilized. Instead, meiosis completion only occurs if fertilization of MII-arrested oocytes takes place. Consider rephrasing to communicate the accurate concept.

- We followed the reviewer's suggestions by rewording (line 54-56 in blue in the new text).

- Lines 145-153 and Figure S4-F: Authors claim that TZP-deprived oocytes grow up to normal sizes. However, the perimeter of fully grown oocytes is lower in *Myo10*^{-/-} full oocytes. This is conflicting.

- Considering this point, two objects of the same size (Fig.2 B) can have a different perimeter (Fig.S4 F) if they do not have the same shape, which is the case here (see aspect ratio Fig.S4 E).

Cross-consultation comments

In addition to the comments made by my own, my colleagues both suggested the inclusion of experiments to determine the functionality of the remaining TZP through dye diffusion assays. I concur with them.

All reviewers asked whether cumulus-oocyte coupling was affected in *Myo10* full knockouts, as they have significantly fewer transzonal projections (TZPs) between the oocyte and the surrounding somatic cells (Fig.1 E). They all proposed dye diffusion assays, *i.e.*, microinjection of a fluorophore into the oocyte to visualize if it spreads into the surrounding somatic cells. We conducted this experiment (see Fig.A below, new Fig.S2 C), injecting Lucifer Yellow into control and *Myo10*^{-/-} full oocytes. In addition, we treated control complexes with carbenoxolone (CBX), a gap-junction blocker, to prevent the transfer of Lucifer Yellow from the oocyte to the surrounding somatic cells via gap-junctions, as a negative control. Our results show that overall, some coupling remains in *Myo10* full knockouts.

Fig.A: Control and *Myo10*^{-/-} full live cumulus-oocyte complexes stained with SiR-actin (magenta) to label TZPs in which each oocyte was injected with 10% Lucifer Yellow (green). Some Control cumulus-oocyte complexes were incubated for 1h in 150 μ M carbenoxolone (CBX-treated) to block gap-junctions. Scale bar 20 μ m.

We then quantified the coupling between the oocyte and follicular cells. For that, we quantified the ratio of Lucifer Yellow fluorescence intensity between the follicular cells and the oocyte 30 min after Lucifer Yellow microinjection into the oocyte. Measurements were performed on control

cumulus-oocyte complexes, *Myo10* full knockouts complexes and control complexes treated with CBX when single follicular cells were visible, still connected to the oocyte by TZPs (visible with SiR-actin). Our results show that the amount of coupling between the oocyte and the follicular cells is comparable between control and *Myo10* full knockouts, and abolished by blocking gap junctions with CBX (see Fig.B below, new Fig.S2 D). Thus, *Myo10* full knockouts have fewer TZPs, but their remaining TZPs are functional, allowing exchanges between the oocyte and the follicular cells, even if globally reduced. This could explain the fact that these oocytes do not escape the prophase arrest.

Fig.B: Scatter plot showing cumulus-oocyte coupling represented by the Lucifer Yellow follicular cell enrichment, quantified by the ratio of Lucifer Yellow fluorescence intensity between the follicular cell and the oocyte 30 min after Lucifer Yellow microinjection into the oocyte. Measurements were performed on cumulus-oocyte complexes when single follicular cells were visible, still connected to the oocyte by TZPs (visible with SiR-actin). Control cumulus-oocyte complexes are in dark gray, *Myo10*^{-/-} full complexes in green, and control complexes incubated for 1h in 150 μ M carbenoxolone (CBX-treated) to block gap-junctions in red. (n) is the number of single follicular cell coupling analyzed. Data are mean \pm s.d. with individual data points plotted. Data are from three independent experiments. Statistical significance of differences was assessed by an ANOVA, P value = 0,1493 (n.s, Ctrl full vs. *Myo10*^{-/-} full), P value < 0,0001 (****, Ctrl full vs. CBX-treated), P value < 0,0001 (****, *Myo10*^{-/-} full vs. CBX-treated), n.s not significant.

Significance (Required):

The manuscript clearly adds to the existing knowledge. I'm convinced that the findings described here will be of interest for readers from the field of reproductive biology, follicle development, and oocyte biology.

Authors are encouraged to better frame their findings as to the existing knowledge. There is at least one another knockout model in mice that leads to TZP density reduction (Zhang et al., 2021; Nature Comm., 12:2523). In this paper, the authors show that the TZPs connecting the GCs and the oocyte support proper oocyte development. Also, its removal results in subfertility. These previous findings should be acknowledged in the current manuscript.

My expertise: researcher in reproductive biology; emphasis on folliculogenesis and oocyte development.

We now cite the following paper (Zhang et al., 2021; Nature Comm., 12:2523) in our discussion as suggested by the reviewer. We did not cite it in the first version of our manuscript because their oocyte-specific *Radixin* knockout causes primarily a loss of oocyte microvilli, inducing an abnormal GC development (including a decrease of TZPs but not only, also affecting GC proliferation and apoptosis), affecting the survival of follicles and shortening the reproductive lifespan in females. The ovaries of oocyte-specific *Radixin* knockouts are smaller, with a lower density of follicles, and a retardation of oocyte growth, consistent with premature ovarian insufficiency, which is not the case for the *Myo10* full knockouts. As such, it is impossible to link subfertility to the decrease of TZPs in the Zhang et al 2021 paper.

Reviewer 2:

Evidence, reproducibility and clarity (Required):

The authors have investigated the effect of knocking out the Myosin-X gene (*Myo10*) on oocytes in mice. The major finding was that transzона processes (TZPs), which are filipodia-like structures that cross the oocyte's extracellular matrix shell (zona pellucida, ZP), were greatly reduced when the gene was globally knocked out. In comparison, an oocyte-specific knockout had no effect on TZPs. Using a machine learning algorithm developed by one of the authors, it was found that characteristics of the ZP were changed, and the oocyte shape was altered in the knockouts. RNAseq showed that many genes were upregulated in oocytes from knockout females. Oocytes from knockouts also failed to complete meiotic maturation at a higher rate and produced embryos that were fertilized less frequently and whose embryos were impaired in reaching the blastocyst stage. Finally, litters per female and pups per litter were lower in knockouts, indicating lower female fertility.

Major comments:

Overall, this is a very well done and comprehensive study that indicates a major role for MYO10 in oogenesis and oocyte developmental competence. There are some relatively major issues that should be resolved, however:

We thank the reviewer for his/her thoughtful comments.

1. An experiment was done to assess the number of follicles per ovary, which is shown in Fig. S3. No significant difference in follicle number (per unit area) was detected. However, there are two problems here. One is that only four repeats were done, and the lack of significance would appear to be driven by only one of the knockout repeats which had a high number of oocytes compared the others. It is possible that there is really not a biologically significant difference between the controls and somatic knockouts, but there are an insufficient number of repeats to determine this (technically, $P > 0.95$ would mean they are the same). Second, it is unclear that the number of follicles per unit area is the relevant parameter for fertility rather than the absolute number of follicles. Both measures should be reported and tested statistically.

The reviewer asked to repeat our transparization experiment with more animals, and we did so. We added 3 mice for each group for a total of 5 mice (10 ovaries) per group, which is above standards (see below, new Fig.S3 B). Our results show that follicular density is not significantly different between control and *Myo10*^{-/-} full ovaries.

Scatter plot of the density of follicles in ovaries (number of follicles per mm²). Control ovaries are in dark gray and *Myo10*^{-/-} full ovaries in green. For each condition, data are from five different mice. (N) is the number of ovaries analyzed. Data are mean±s.d. with individual data points plotted. Statistical significance of differences was assessed by a two-tailed unpaired t test with Welch's correction, P value = 0.9389. n.s not significant.

Concerning the relevant parameter for fertility, we do believe that the number of follicles per unit area is the relevant one, since it overcomes the physiological variability in ovarian size. We have plotted below the weight of the ovaries of each animal (24 control mice and 27 *Myo10* full knockouts), light and dark grey for the ovaries of the same control mouse, light and dark green for the ovaries of the same *Myo10* full knockout. It is very clear that in most cases, the weight of the ovaries is very variable in the same mouse, and can go from simple to double.

2. A main function of TZPs is to transfer metabolites and other small molecules into the oocyte via Cx37-containing gap junctions. As the authors note, the phenotype here is different from the Cx37 knockout, where oocytes failed to develop. This implies some connectivity remains in *Myo10* knockouts, but how much has not been determined. The amount of connectivity should be measured. The techniques are fairly straightforward and involve only microinjection of a fluorophore into the oocyte and measuring the spread into the surrounding somatic cells. This also has implications for the lack of effect on GVBD and resumption of meiosis, since Laurinda Jaffe's group has shown that diffusion of cGMP out through the gap junctions is important in this process.

All reviewers asked whether cumulus-oocyte coupling was affected in *Myo10* full knockouts, as they have significantly fewer transzonal projections (TZPs) between the oocyte and the surrounding somatic cells (Fig.1 E). They all proposed dye diffusion assays, *i.e.*, microinjection of a fluorophore into the oocyte to visualize if it spreads into the surrounding somatic cells. We conducted this experiment (see Fig.A below, new Fig.S2 C), injecting Lucifer Yellow into control and *Myo10*^{-/-} full oocytes. In addition, we treated control complexes with carbenoxolone (CBX), a gap-junction blocker, to prevent the transfer of Lucifer Yellow from the oocyte to the surrounding somatic cells via gap-junctions, as a negative control. Our results show that overall, some coupling remains in *Myo10* full knockouts.

Fig.A: Control and *Myo10*^{-/-} full live cumulus-oocyte complexes stained with SiR-actin (magenta) to label TZPs in which each oocyte was injected with 10% Lucifer Yellow (green). Some control cumulus-oocyte complexes were incubated for 1h in 150 μ M carbenoxolone (CBX-treated) to block gap-junctions. Scale bar 20 μ m.

We then quantified the coupling between the oocyte and follicular cells. For that, we quantified the ratio of Lucifer Yellow fluorescence intensity between the follicular cells and the oocyte 30 min after Lucifer Yellow microinjection into the oocyte. Measurements were performed on control cumulus-oocyte complexes, *Myo10* full knockouts complexes and control complexes treated with CBX when single follicular cells were visible, still connected to the oocyte by TZPs (visible with SiR-actin). Our results show that the amount of coupling between the oocyte and the follicular cells is comparable between control and *Myo10* full knockouts, and abolished by blocking gap junctions with CBX (see Fig.B below, new Fig.S2 D). Thus, *Myo10* full knockouts have fewer TZPs, but their remaining TZPs are functional, allowing exchanges between the oocyte and the follicular cells, even if globally reduced. This could explain the fact that these oocytes do not escape the prophase arrest.

Fig.B: Scatter plot showing cumulus-oocyte coupling represented by the Lucifer Yellow follicular cell enrichment, quantified by the ratio of Lucifer Yellow fluorescence intensity between the follicular cell and the oocyte 30 min after Lucifer Yellow microinjection into the oocyte. Measurements were performed on cumulus-oocyte complexes when single follicular cells were visible, still connected to the oocyte by TZPs (visible with SiR-actin). Control cumulus-oocyte complexes are in dark gray, *Myo10*^{-/-} full complexes in green, and control complexes incubated for 1h in 150 μ M carbenoxolone (CBX-treated) to block gap-junctions in red. (n) is the number of single follicular cell coupling analyzed. Data are mean \pm s.d. with individual data points plotted. Data are from three independent experiments. Statistical significance of differences was assessed by an ANOVA, P value = 0,1493 (n.s, Ctrl full vs. *Myo10*^{-/-} full), P value < 0,0001 (****, Ctrl full vs. CBX-treated), P value < 0,0001 (****, *Myo10*^{-/-} full vs. CBX-treated), n.s not significant.

3. The TZP-like structures that remain are intriguing, but this was not followed up. They apparently are visible optically but contain neither actin nor membrane. Is it possible that these are tracks left

from degenerated TZPs? Electron microscopy might resolve this question and should be considered. In any case, a more extensive discussion is warranted since the data are contradictory, with fluorescence-based methods indicating a decrease in TZPs but optical methods indicating an apparent increase.

All reviewers asked why there is an apparent discrepancy between the results for detecting the numbers and densities of TZPs. This is because we did not explain the phenotype well enough and we apologize for this. There is in fact no contradiction between the number and density of TZPs in *Myo10* full knockouts in the manuscript. The TZP-like structure that are visible in transmitted light in the *Myo10* full knockouts always contain actin (see Fig.C below). We see less TZPs in *Myo10* full knockouts with fluorescence-based methods (Fig.1 E), but most of these remaining TZPs are visible by transmitted light. In control oocytes, TZPs are abundant, as evidenced with fluorescence-based methods (Fig.1 E), but not visible by transmitted light (Fig.2 F and Fig.C below). These data were missing from our initial submission and we added them in our revised manuscript (new Fig.2 F).

Fig.C: Cropped images of *zona pellucida* from control and *Myo10*^{-/-} full oocytes. Brightfield images on the left panels, SiR-actin (white) labeled TZPs on the middle panels, merge (SiR-actin in magenta) on the right panels. Scale bar 9 μ m.

Our observations suggest that in *Myo10* full knockouts, the structure of the remaining TZPs and/or that of the *zona pellucida* is different from that of control oocytes. Interestingly, we occasionally

observe some follicular-like cells located ectopically within the perivitelline space of *Myo10* full knockouts (Fig.S4 C), and the texture of the *zona pellucida* is different between control and *Myo10* full knockouts (Fig.2 C). We therefore hypothesize that the structure of the *zona pellucida* (a viscoelastic extracellular matrix) may change as a function of the crosslinking generated by the number of TZPs passing through it. If so, this could change its mechanical properties. Reviewer 2 proposed to perform electron microscopy to study these TZP-like structures that he thought lacked actin. Since this is not the case (see Fig.C above), and since our hypothesis is that the *zona pellucida* might have different properties, we conducted experiments to assess the mechanical properties of the *zona pellucida* of control and *Myo10* full knockouts using Atomic Force Microscopy (AFM). Our experiments show a decreased *zona pellucida* elasticity in *Myo10* full knockouts (see Fig.D below, new Fig.S4 H). This suggests that the reduced number of TZPs could affect the structure, maybe via reduced crosslinking of the *zona pellucida*, and thus the mechanical properties of the *zona pellucida*. We discuss this point in the discussion.

Fig.D: Elasticity of the *zona pellucida* of control and *Myo10*^{-/-} full oocytes, measured with Atomic Force Microscopy (AFM). Data are mean±s.d. with individual data points plotted. Data are from three independent experiments. Statistical significance of differences was assessed by a two-tailed Mann Whitney test, P value = 0.0098.

4. The apparent delay in formation of a perivitelline space is interesting. The perivitelline space forms gradually as the ZP detaches from the oocyte independent of meiotic maturation (see, e.g., Richard et al., 2017, J Cell Physiol 232:2436-46). Could this not be a delay in detachment and therefore transient (and dependent on when the assay was performed relative to oocyte isolation)?

We are not sure to fully understand the point of the reviewer. Perivitelline space formation is not delayed but rather precocious and more homogeneous in *Myo10* full knockouts, see Fig.2 E.

5. While GO analysis was done and shown in Table 1, this is not treated in any depth in the paper. There should be more description of the GO pathways that were upregulated and the implications.

In our discussion, we comment directly on the most deregulated transcripts in TZP-deprived oocytes. These transcripts are involved in the oxidative stress response (*Oser1*), the metabolism of polyamine (*Akr1b3*, *Oaz1*) and cell cycle transitions (*Psm5*, *Cyclin B1*, Fig.3 C, Table1). These transcripts do belong to some of the biological process pathways found in our GO analysis (polyamine metabolic process, negative regulation of reactive oxygen species biosynthesis process). These pathways when deregulated are known to affect subsequent meiotic maturation and early embryogenesis (see references in the discussion). However, we prefer not to discuss the GO further, as it seems too speculative.

Minor comments:

1. The comparisons that were done for whole-body knockout vs. oocyte-specific knockouts were only done by comparing each to its control. There is no direct comparison showing whether the two knockouts differ significantly from each other. The comparisons should be done using ANOVA with appropriate post-hoc tests to test all four groups against each other.

We made the comparisons between mouse strains using ANOVA as suggested by the reviewer for Fig.1 E, Fig.2 B, Fig.2 C, Fig.2 E, Fig.2 G, Fig.S4 A, Fig.S4 B, Fig.S4 E, Fig.S4 F (see below, old graphs paired with new statistics) and find, as expected, that all controls and the oocyte-specific knockout are not significantly different from each other, whereas the whole-body knockout is significantly different from all of them. We included these new statistics in our revised manuscript.

ANOVA tests comparing the mean of each column with the mean of every other column when the data followed a Gaussian distribution, and Kruskal-Wallis tests comparing the mean rank of each column with the mean rank of every other column when the data did not.

Fig.1 E

Ctrl full vs. Myo10 ^{-/-} full	****	<0,0001
Ctrl full vs. Ctrl oocyte	ns	0,6857
Ctrl full vs. Myo10 ^{-/-} oocyte	ns	0,4479
Myo10 ^{-/-} full vs. Ctrl oocyte	****	<0,0001
Myo10 ^{-/-} full vs. Myo10 ^{-/-} oocyte	****	<0,0001
Ctrl oocyte vs. Myo10 ^{-/-} oocyte	ns	0,9722

Fig.2 B

Ctrl full vs. Myo10 ^{-/-} full	ns	0,4459
Ctrl full vs. Ctrl oocyte	ns	0,7355
Ctrl full vs. Myo10 ^{-/-} oocyte	ns	>0,9999
Myo10 ^{-/-} full vs. Ctrl oocyte	ns	0,9905
Myo10 ^{-/-} full vs. Myo10 ^{-/-} oocyte	ns	>0,9999
Ctrl oocyte vs. Myo10 ^{-/-} oocyte	ns	>0,9999

Fig.2 C

Ctrl full vs. Myo10 ^{-/-} full	*	0,0280
Ctrl full vs. Ctrl oocyte	ns	0,3355
Ctrl full vs. Myo10 ^{-/-} oocyte	ns	>0,9999
Myo10 ^{-/-} full vs. Ctrl oocyte	****	<0,0001
Myo10 ^{-/-} full vs. Myo10 ^{-/-} oocyte	**	0,0083
Ctrl oocyte vs. Myo10 ^{-/-} oocyte	ns	0,3895

Ctrl full vs. Myo10 ^{-/-} full	****	<0,0001
Ctrl full vs. Ctrl oocyte	ns	>0,9999
Ctrl full vs. Myo10 ^{-/-} oocyte	ns	0,3139
Myo10 ^{-/-} full vs. Ctrl oocyte	****	<0,0001
Myo10 ^{-/-} full vs. Myo10 ^{-/-} oocyte	*	0,0103
Ctrl oocyte vs. Myo10 ^{-/-} oocyte	ns	0,0811

Ctrl full vs. Myo10 ^{-/-} full	****	<0,0001
Ctrl full vs. Ctrl oocyte	ns	0,7183
Ctrl full vs. Myo10 ^{-/-} oocyte	ns	0,7291
Myo10 ^{-/-} full vs. Ctrl oocyte	****	<0,0001
Myo10 ^{-/-} full vs. Myo10 ^{-/-} oocyte	****	<0,0001
Ctrl oocyte vs. Myo10 ^{-/-} oocyte	ns	0,9999

Ctrl full vs. Myo10 ^{-/-} full	ns	>0,9999
Ctrl full vs. Ctrl oocyte	ns	0,0503
Ctrl full vs. Myo10 ^{-/-} oocyte	ns	0,0566
Myo10 ^{-/-} full vs. Ctrl oocyte	**	0,0063
Myo10 ^{-/-} full vs. Myo10 ^{-/-} oocyte	*	0,0388
Ctrl oocyte vs. Myo10 ^{-/-} oocyte	ns	>0,9999

Ctrl full vs. Myo10 ^{-/-} full	ns	>0,9999
Ctrl full vs. Ctrl oocyte	ns	0,9997
Ctrl full vs. Myo10 ^{-/-} oocyte	ns	0,5007
Myo10 ^{-/-} full vs. Ctrl oocyte	ns	0,9941
Myo10 ^{-/-} full vs. Myo10 ^{-/-} oocyte	ns	0,9522
Ctrl oocyte vs. Myo10 ^{-/-} oocyte	ns	0,8581

Fig.S4 E

Ctrl full vs. Myo10 ^{-/-} full	**	0,0030
Ctrl full vs. Ctrl oocyte	ns	0,4773
Ctrl full vs. Myo10 ^{-/-} oocyte	ns	0,4353
Myo10 ^{-/-} full vs. Ctrl oocyte	ns	>0,9999
Myo10 ^{-/-} full vs. Myo10 ^{-/-} oocyte	ns	>0,9999
Ctrl oocyte vs. Myo10 ^{-/-} oocyte	ns	>0,9999

Fig.S4 F

Ctrl full vs. Myo10 ^{-/-} full	*	0,0412
Ctrl full vs. Ctrl oocyte	ns	0,6078
Ctrl full vs. Myo10 ^{-/-} oocyte	ns	>0,9999
Myo10 ^{-/-} full vs. Ctrl oocyte	ns	0,9516
Myo10 ^{-/-} full vs. Myo10 ^{-/-} oocyte	ns	>0,9999
Ctrl oocyte vs. Myo10 ^{-/-} oocyte	ns	>0,9999

2. The experiment in which 5-ethynyl uridine incorporation was used to show that global transcription was not increased may not actually be conclusive, since a large amount of RNA synthesized is not mRNA. A global increase in mRNA synthesis could still be occurring but the signal swamped by RNAs such as rRNA and other non-coding RNAs.

We agree with the reviewer, and that is why we have been careful to talk only about global RNA synthesis (and not mRNA synthesis), and not to draw too strong conclusions from this experiment in our initial manuscript.

Cross-consultation comments

It looks like the reviewers basically agree that this is interesting but there are questions remaining about whether cumulus-oocyte coupling is affected (and could explain the phenotype) and why there is an apparent discrepancy between the results for detecting the numbers and densities of TZPs. These should be addressed.

These points were addressed (see before).

Significance (Required):

This work has fundamental implications for understanding oocyte development and the role of the surrounding somatic cells in oogenesis and oocyte developmental competence. It also has direct implications for human and animal fertility and assisted reproduction.

This is a fundamental new set of results that establishes a role for Myo10 and adds to the knowledge about the role of transzonal processes. It is a substantial advance over previously published research.

The audience will primarily be basic biomedical researchers in the general field of reproductive biology as well as those investigating filipodia and should extend to those interested in translational research in infertility.

I have direct and extensive expertise in the field of oogenesis in mice.

Reviewer 3:

Evidence, reproducibility and clarity (Required):

The objective of this manuscript was to determine the role of TZPs in mouse oocyte quality. The experimental plan was to compare the phenotypes of global Myo10^{-/-}, oocyte Myo10^{-/-}, and Myo10^{+/+} follicles. The results indicate that global loss of Myo10 did not prevent oocyte growth, but resulted in lower density of TZPS. Whole ovary image analysis revealed that Myo10^{-/-} follicles actually contained more TZPs than wt, despite the fact that TZP density was decreased in Myo10^{-/-} follicles. In mature knockout females, oocyte growth proceeded, but with impaired oocyte-zona integrity and alterations in gene expression including upregulation of numerous protein encoding genes. Oocytes from Myo10^{-/-} knockout females produced a normal-appearing spindle but exhibited reduced capacity to mature beyond MI. Analysis of ovulated oocytes from mated females revealed an increase in the number of unfertilized and dead oocytes, many of which exhibited gaps between the zona pellucida and the oocyte plasma membrane. Those oocytes that were successfully fertilized exhibited a higher than normal of developmental arrest by the blastocyst stage. Lastly, mating trials revealed that Myo10^{-/-} females were sub-fertile.

The results are clearly described with high quality imaging to demonstrate phenotypes. The data appear reproducible based on sample size and the number of repetitions. In most cases, statistical analysis demonstrates significance of observed differences.

We thank the reviewer for his/her supporting comments.

Minor comments:

1. Fig. 2B does not provide statistical evidence that the two data sets differ.

In Fig.2 B, the data sets do not differ.

Fig.2 B

Ctrl full vs. Myo10 ^{-/-} full	ns	0,4459
Ctrl full vs. Ctrl oocyte	ns	0,7355
Ctrl full vs. Myo10 ^{-/-} oocyte	ns	>0,9999
Myo10 ^{-/-} full vs. Ctrl oocyte	ns	0,9905
Myo10 ^{-/-} full vs. Myo10 ^{-/-} oocyte	ns	>0,9999
Ctrl oocyte vs. Myo10 ^{-/-} oocyte	ns	>0,9999

2. Fig. 6A Was the zona pellucida functional in unfertilized oocytes from *Myo10*^{-/-} females? That is, were sperm bound to the zona or within the perivitelline space?

We will not *in vitro* fertilize superovulated *Myo10* full knockouts to check if sperm is bound to the *zona pellucida* or within the perivitelline space, since our *in vivo* matings (Fig.6 A) show that *Myo10* full knockouts can be fertilized, albeit less efficiently than controls, suggesting that the *zona pellucida* is at least partially functional. While we recognize that this is an interesting question, we believe that it could be the object of a future study, more focused on the *zona pellucida*.

3. The observation that oocytes from *Myo10*^{-/-} females have more TZPs but lower TZP density raises questions as to how more TZPs (even if less densely spaced) could fail to support oocyte development. Dye diffusion assays comparing the rate of injected dye from *Myo10*^{+/+} and *Myo10*^{-/-} (GV stage) or (maturing) stage oocytes into their attached granulosa cells might reveal an explanation.

All reviewers asked whether cumulus-oocyte coupling was affected in *Myo10* full knockouts, as they have significantly fewer transzonal projections (TZPs) between the oocyte and the surrounding somatic cells (Fig.1 E). They all proposed dye diffusion assays, *i.e.*, microinjection of a fluorophore into the oocyte to visualize if it spreads into the surrounding somatic cells. We conducted this experiment (see Fig.A below, new Fig.S2 C), injecting Lucifer Yellow into control and *Myo10*^{-/-} full oocytes. In addition, we treated control complexes with carbenoxolone (CBX), a gap-junction blocker, to prevent the transfer of Lucifer Yellow from the oocyte to the surrounding somatic cells via gap-junctions, as a negative control. Our results show that overall, some coupling remains in *Myo10* full knockouts.

Fig.A: Control and *Myo10*^{-/-} full live cumulus-oocyte complexes stained with SiR-actin (magenta) to label TZPs in which each oocyte was injected with 10% Lucifer Yellow (green). Some control cumulus-oocyte complexes were incubated for 1h in 150 μ M carbenoxolone (CBX-treated) to block gap-junctions. Scale bar 20 μ m.

We then quantified the coupling between the oocyte and follicular cells. For that, we quantified the ratio of Lucifer Yellow fluorescence intensity between the follicular cells and the oocyte 30 min after Lucifer Yellow microinjection into the oocyte. Measurements were performed on control

cumulus-oocyte complexes, *Myo10* full knockouts complexes and control complexes treated with CBX when single follicular cells were visible, still connected to the oocyte by TZPs (visible with SiR-actin). Our results show that the amount of coupling between the oocyte and the follicular cells is comparable between control and *Myo10* full knockouts, and abolished by blocking gap junctions with CBX (see Fig.B below, new Fig.S2 D). Thus, *Myo10* full knockouts have fewer TZPs, but their remaining TZPs are functional, allowing exchanges between the oocyte and the follicular cells, even if globally reduced. This could explain the fact that these oocytes do not escape the prophase arrest.

Fig.B: Scatter plot showing cumulus-oocyte coupling represented by the Lucifer Yellow follicular cell enrichment, quantified by the ratio of Lucifer Yellow fluorescence intensity between the follicular cell and the oocyte 30 min after Lucifer Yellow microinjection into the oocyte. Measurements were performed on cumulus-oocyte complexes when single follicular cells were visible, still connected to the oocyte by TZPs (visible with SiR-actin). Control cumulus-oocyte complexes are in dark gray, *Myo10*^{-/-} full complexes in green, and control complexes incubated for 1h in 150 μ M carbenoxolone (CBX-treated) to block gap-junctions in red. (n) is the number of single follicular cell coupling analyzed. Data are mean \pm s.d. with individual data points plotted. Data are from three independent experiments. Statistical significance of differences was assessed by an ANOVA, P value = 0,1493 (n.s, Ctrl full vs. *Myo10*^{-/-} full), P value < 0,0001 (****, Ctrl full vs. CBX-treated), P value < 0,0001 (****, *Myo10*^{-/-} full vs. CBX-treated), n.s not significant.

February 27, 2023

RE: Life Science Alliance Manuscript #LSA-2023-01963-TR

Dr. Marie-Emilie Terret
Center for Interdisciplinary Research in Biology
CIRB
11 place Marcelin Berthelot
Paris 75005
France

Dear Dr. Terret,

Thank you for submitting your revised manuscript entitled "filopodia-like protrusions of adjacent somatic cells shape the developmental potential of oocytes". We would be happy to publish your paper in Life Science Alliance pending final revisions necessary to meet our formatting guidelines.

- please refer to the remaining Reviewer comments
- please provide your manuscript text as an editable doc file
- please provide any table files as editable doc or excel files
- please add a Data Availability Statement to your Materials and Methods section to mention the RNA-seq accession information and Galaxy workflow link again

A. FINAL FILES:

B. MANUSCRIPT ORGANIZATION AND FORMATTING:

**Submission of a paper that does not conform to Life Science Alliance guidelines will delay the acceptance of your

manuscript.**

The license to publish form must be signed before your manuscript can be sent to production. A link to the electronic license to publish form will be sent to the corresponding author only. Please take a moment to check your funder requirements.

Sincerely,

Reviewer #1 (Comments to the Authors (Required)):

1. Short summary:

The short summary is unchanged since last review.

2. Main points:

The authors have adequately addressed previous comments and questions. In particular, the new dye-coupling experiments are informative, and the results are unexpected and interesting. This resolved the question of whether the phenotype was the result of decreased coupling or, as shown, other properties of the transzonal processes.

For my previous comment 4, I was not clear, and meant that the formation of the perivitelline space may be delayed in the controls compared to the KOs. The question is whether the observation reflects a reduced incidence of PVS formation or a delay in PVS formation that is captured as a difference at the single timepoint. It is up to the authors whether to address this or not in the manuscript, as it is not a critical point.

3. Additional issues:

No additional issues remain.

Reviewer #2 (Comments to the Authors (Required)):

The objective of this manuscript was to determine the role of TZPs in mouse oocyte quality. The work presented here substantially improves our understanding of TZP structure and function, revealing novel aspects of follicle development that open up a new level of interest in oocyte development through the use on state of the art methodology. The experimental plan was to compare the phenotypes of global Myo10^{-/-}, oocyte Myo10^{-/-}, and Myo10^{+/+} follicles. The results indicate that global loss of Myo10 did not prevent oocyte growth, but did lead to lower density of TZPS, changes in gene expression, reduced oocyte quality and developmental competence. The issues identified in the initial submission have now been fully satisfied and I believe that the manuscript is now ready for publication.

Cross comments among reviews: I had cross comments on two aspects of the manuscript, the dye diffusion experiment and the TZP abundance imaged by fluorescence vs transmitted light.

Reviewer #3 (Comments to the Authors (Required)):

I have reviewed this manuscript (Crozet et al. - LSA-2023-01963-TR) previously, and my positive view of the manuscript

remains.

In addition to the major findings already described in the first round of review (decrease in TZP density, discrete morphological alterations in the oocytes, alterations in oocyte gene expression, the inability of the oocytes to complete the first meiotic division - lack of 1PB extrusion, and subfertility in *Myo10*^{-/-} full females) the authors now included an assessment of the functional coupling between oocyte and the surrounding somatic cells. Using Luciferase Yellow diffusion, they demonstrated that the remaining TZPs (in *Myo10*^{-/-} full females) are functional and allow the diffusion of the dye similarly to control. This partially explains the phenotypes in *Myo10* knockouts.

Effects of *Myo10* knockout in mechanical properties were also included in this version.

The manuscript clearly advances the field and the results described justifies its publication. I recommend the acceptance.

Minor

Lines 844 and 1208: check the concentration of Lucifer Yellow (5% or 10%?)

We thank the three reviewers for their clearly supportive appreciation of our revised manuscript.

Reviewer #1 (Comments to the Authors (Required)):

1. Short summary:

The short summary is unchanged since last review.

No answer required.

2. Main points:

The authors have adequately addressed previous comments and questions. In particular, the new dye-coupling experiments are informative, and the results are unexpected and interesting. This resolved the question of whether the phenotype was the result of decreased coupling or, as shown, other properties of the transzonal processes.

No answer required.

For my previous comment 4, I was not clear, and meant that the formation of the perivitelline space may be delayed in the controls compared to the KOs. The question is whether the observation reflects a reduced incidence of PVS formation or a delay in PVS formation that is captured as a difference at the single timepoint. It is up to the authors whether to address this or not in the manuscript, as it is not a critical point.

We thank the reviewer for this clarification. We prefer not to address this since we have no experiment allowing to discriminate the two hypothesis.

3. Additional issues:

No additional issues remain.

No answer required.

Reviewer #2 (Comments to the Authors (Required)):

The objective of this manuscript was to determine the role of TZPs in mouse oocyte quality. The work presented here substantially improves our understanding of TZP structure and function, revealing novel aspects of follicle development that open up a new level of interest in oocyte development through the use on state of the art methodology. The experimental plan was to compare the phenotypes of global Myo10^{-/-}, oocyte Myo10^{-/-}, and Myo10^{+/+} follicles. The results indicate that global loss of Myo10 did not prevent oocyte growth, but did lead to lower density of TZPS, changes in gene expression, reduced oocyte quality and developmental competence. The issues identified in the initial submission have now been fully satisfied and I believe that the manuscript is now ready for publication.

No answer required.

Cross comments among reviews: I had cross comments on two aspects of the manuscript, the dye diffusion experiment and the TZP abundance imaged by fluorescence vs transmitted light.

No answer required.

Reviewer #3 (Comments to the Authors (Required)):

I have reviewed this manuscript (Crozet et al. - LSA-2023-01963-TR) previously, and my positive view of the manuscript remains.

In addition to the major findings already described in the first round of review (decrease in TZP density, discrete morphological alterations in the oocytes, alterations in oocyte gene expression, the inability of the oocytes to complete the first meiotic division - lack of 1PB extrusion, and subfertility in Myo10^{-/-} full females) the authors now included an assessment of the functional coupling between oocyte and the surrounding somatic cells. Using Luciferase Yellow diffusion, they demonstrated that the remaining TZPs (in Myo10^{-/-} full females) are functional and allow the diffusion of the dye similarly to control. This partially explains the phenotypes in Myo10 knockouts.

Effects of Myo10 knockout in mechanical properties were also included in this version.

The manuscript clearly advances the field and the results described justifies its publication. I recommend the acceptance.

No answer required.

Minor

Lines 844 and 1208: check the concentration of Lucifer Yellow (5% or 10%?)

We thank the reviewer for pointing out this discrepancy. The concentration is 10%, we have changed it on line 844.

March 7, 2023

RE: Life Science Alliance Manuscript #LSA-2023-01963-TRR

Dr. Marie-Emilie Terret
Center for Interdisciplinary Research in Biology
CIRB
11 place Marcelin Berthelot
Paris 75005
France

Dear Dr. Terret,

Thank you for submitting your Research Article entitled "filopodia-like protrusions of adjacent somatic cells shape the developmental potential of oocytes". It is a pleasure to let you know that your manuscript is now accepted for publication in Life Science Alliance. Congratulations on this interesting work.

DISTRIBUTION OF MATERIALS:

Again, congratulations on a very nice paper. I hope you found the review process to be constructive and are pleased with how the manuscript was handled editorially. We look forward to future exciting submissions from your lab.

Sincerely,
